# Microbiome-mediated fructose depletion restricts murine gut colonization by vancomycin-resistant *Enterococcus*

Sandrine Isaac [1,7,8] ✉, Alejandra Flor-Duro[1,8], Gloria Carruana[1], Leonor Puchades-Carrasco [2], Anna Quirant[1], Marina Lopez-Nogueroles[3], Antonio Pineda-Lucena[2,4], Marc Garcia-Garcera [5] & Carles Ubeda [1,6] ✉

Multidrug-resistant organisms (MDRO) are a major threat to public health. MDRO infections, including those caused by vancomycin-resistant *Enterococcus* (VRE), frequently begin by colonization of the intestinal tract, a crucial step that is impaired by the intestinal microbiota. However, the specific members of the microbiota that suppress MDRO colonization and the mechanisms of such protection are largely unknown. Here, using metagenomics and mouse models that mimic the patients' exposure to antibiotics, we identified commensal bacteria associated with protection against VRE colonization. We further found a consortium of five strains that was sufficient to restrict VRE gut colonization in antibiotic treated mice. Transcriptomics in combination with targeted metabolomics and in vivo assays indicated that the bacterial consortium inhibits VRE growth through nutrient depletion, specifically by reducing the levels of fructose, a carbohydrate that boosts VRE growth in vivo. Finally, in vivo RNA-seq analysis of each strain of the consortium in combination with ex vivo and in vivo assays demonstrated that a single bacterium (*Olsenella sp.*) could recapitulate the effect of the consortium. Our results indicate that nutrient depletion by specific commensals can reduce VRE intestinal colonization, which represents a novel non-antibiotic based strategy to prevent infections caused by this multidrug-resistant organism.

Multidrug resistant organisms (MDRO), including vancomycin-resistant *Enterococcus* (VRE), have become a major threat to public health, compromising our capacity to treat common infectious diseases and increasing the risk of hospital medical procedures. *Enterococcus* isolates, including VRE, are among the third- to fourth-most prevalent pathogens causing infections in hospitalized patients worldwide, and can potentially lead to lethal outcomes[1,2]. Because of its wide resistance to multiple antibiotics, which impairs its treatment[3], VRE is listed among top priority multidrug-resistant pathogens for which new therapies should be developed according to the World Health Organization[4]. Consequently, novel antibiotics have been introduced to treat VRE infections[3]. However, strains resistant to these new antimicrobials rapidly emerge[2,3], which encourage the implementation of non-antibiotic strategies to prevent infections by this clinically challenging pathogen.

[1]Fundación para el Fomento de la Investigación Sanitaria y Biomédica de la Comunitat Valenciana - FISABIO, Valencia, Spain. [2]Drug Discovery Unit, Instituto de Investigación Sanitaria La Fe, Valencia, Spain. [3]Analytical Unit Platform, Instituto de Investigación Sanitaria La Fe, Valencia, Spain. [4]Molecular Therapeutics Program, Centro de Investigación Médica Aplicada, University of Navarra, Pamplona, Spain. [5]Department of Fundamental Microbiology, University of Lausanne, Lausanne, Switzerland. [6]CIBER en Epidemiología y Salud Pública, Madrid, Spain. [7]Present address: Laboratory of Molecular Microbiology, Global Health Institute, School of Life Sciences, Ecole Polytechnique Fédérale de Lausanne, Lausanne, Switzerland. [8]These authors contributed equally: Sandrine Isaac, Alejandra Flor-Duro. ✉e-mail: isaacsandrine8488@gmail.com; ubeda_carmor@gva.es

VRE can colonize hospitalized patients through contamination of skin wounds or catheters, which can lead to urinary tract infections or bacteraemia[5]. In addition, VRE infections can frequently start by the colonization of the intestinal tract[6], a crucial step that is suppressed by commensal microbes inhabiting the gut (i.e. the microbiota)[7]. However, antibiotic therapies disrupt the microbiota, enabling VRE to colonize the intestinal tract to extremely high levels[7–10]. Subsequently, VRE dense colonization of the gut promotes its dissemination to the bloodstream[6,7], and to other patients through faecal contamination of the environment[9,11]. Thus, the microbiota acts as a natural defence that prevents most of the potential infections that could be caused by VRE, while antibiotics enhance infections by allowing the first step of VRE gut colonization. Despite its clinical relevance, only a few studies have started to define the members of the microbiota that are key for conferring protection against VRE and the mechanisms providing such protection. Specifically, two studies showed that production of inhibitory molecules by specific commensal bacteria restricts VRE gut colonization[12,13]. *Blautia producta*, an anaerobic commensal bacterium, produces a lantibiotic that directly inhibits VRE growth in mice and in vitro[8,12]. Moreover, the presence of this lantibiotic is associated with lower VRE faecal levels in humans[12]. Similarly, *E. faecalis* containing a conjugative plasmid expressing a bacteriocin clears VRE gut colonization in mice[13].

Besides production of inhibitory molecules, recent insights into the complex relationships between bacterial species suggests that, by competing for nutrients that are necessary for pathogen growth, commensal bacteria could confer resistance against gut colonization by opportunistic pathogens[14]. This mechanism has been shown to confer colonization resistance to species from the family *Enterobacteriaceae*[15,16]. With respect to VRE, an attempt to identify such mechanism was performed using ex vivo assays[17]. However, results from this particular study could not establish a relation between nutrient depletion by commensal microbes and VRE growth and concluded that inhibitory molecules produced by the microbiota rather than nutrient competition play a role against VRE colonization. More recently, a study investigating the impact of *Enterococcus* on graft-versus-host disease demonstrated that depletion from the diet of a single carbohydrate (i.e. lactose) diminish *Enterococcus* gut levels[18]. Consistent with a major effect of sugar availability on *Enterococcus* gut colonization, the genetic acquisition by *Enterococcus* of specific phosphotransferase systems (PTS), involved in the uptake of simple sugars that are frequently found in the gut, seems to be a major driver for the evolution and emergence of enterococcal clinical isolates[19]. Thus, nutrient availability, and more specifically sugars, could be decisive for VRE gut colonization and, presumably, their depletion by commensal microbes could be a key unidentified mechanism by which the microbiota confers protection. Although this possibility remains to be elucidated, if certain, it could provide new strategies (non-antibiotic based) to prevent infections caused by this multidrug-resistant pathogen.

Here, using a mouse model and different omic techniques (i.e., metagenomics, metatranscriptomics and targeted metabolomics), we have identified commensal bacteria required for restricting VRE intestinal colonization and the mechanism of such inhibition. We found that inoculating a consortium of 5 commensal bacterial isolates (i.e., *Alistipes, Barnesiella, Olsenella, Oscillibacter* and *Flavonifractor*) was sufficient to restrict VRE intestinal colonization in antibiotic treated mice. Moreover, we demonstrated that reduction of VRE levels was driven by diminishing the availability of fructose, a sugar frequently found in the diet that promotes VRE growth in vitro and in vivo. Finally, transcriptomic analysis in combination with ex vivo and in vivo experiments showed that the effect of the consortium could be recapitulated by a single bacterium (i.e., *Olsenella* sp.). Our results have identified a novel bacterium that decreases VRE gut colonization and demonstrated a mechanism by which the commensal microbiota

restricts VRE intestinal colonization through nutrient competition. These results open new non-antibiotic strategies to prevent infections produced by a pathogen that has acquired resistance to most available antibiotics.

## Results

### Different antibiotics induce distinct dysbiotic states and grades of susceptibility to VRE intestinal colonization

We have previously demonstrated that specific antibiotics, including ampicillin or vancomycin, alter the microbiome, which enables VRE intestinal colonization in mice[7,10,20]. To better understand how microbiome changes induced by antibiotics promote VRE intestinal colonization, we treated mice for one week with antibiotics of different spectrum (i.e. ciprofloxacin, neomycin, ceftriaxone, ampicillin, clindamycin and vancomycin, Fig. 1a, Supplementary Fig. 1, methods). Subsequently, mice were inoculated through oral gavage with VRE (strain ATCC700221, used in previous studies to investigate VRE gut colonization in mice[7,10,20]). Mice were housed individually to avoid microbiome transmission among mice sharing the same cage. Immediately before VRE inoculation, faecal pellets were collected for microbiome composition analysis (see methods). Subsequently, the intestinal levels of VRE were analysed 2 days after VRE inoculation (model of antibiotic treatment without recovery, Fig. 1a). All mice that received antibiotics had lower microbiota diversity as compared to untreated mice (Supplementary Fig. 2A), although the effect was greater for those mice that received vancomycin and clindamycin and non-significant for those that received ciprofloxacin. Similarly, a lower microbiota richness (i.e. number of identified Operational Taxonomical Units – OTUs) and faecal biomass (ng of DNA/g of faeces) was detected in those mice that received antibiotics as compared to untreated mice (Supplementary Fig. 2B, C). We next analysed overall microbiota changes by applying NMDS analysis to Bray-Curtis distances obtained among pairs of samples. Mice clustered in this analysis based on the antibiotics they received (Fig. 1b), indicating that specific antibiotics induce reproducible microbiota changes. As seen for the alpha diversity, beta diversity was greatly affected by vancomycin and clindamycin treatment, while other antibiotics such as neomycin had a minor effect (Fig. 1b). In concordance with the alpha and beta diversity changes, vancomycin and clindamycin were also the antibiotics that induced significant changes in a higher number of taxa and OTUs (Fig. 1c, Supplementary Fig. 3, Supplementary Data File 1, Supplementary Data File 2). We next evaluated the effect of antibiotic treatment on the capacity of VRE to colonize the intestinal tract. All antibiotics promoted VRE intestinal colonization as compared to untreated mice that were almost completely resistant (Fig. 1d), although there were marked differences depending on the antibiotic. Consistent with the microbiota analysis, vancomycin and clindamycin were among those antibiotics that allowed higher levels of VRE intestinal colonization (>$10^8$ CFUs / 100 mg of faeces), while neomycin or ciprofloxacin allowed VRE intestinal colonization to a lower extent. Similar results were obtained in analysis performed in caecal samples (Supplementary Fig. 4). Notably, although ampicillin induced less microbiota changes as compared to vancomycin and clindamycin (Fig. 1c, Supplementary Fig. 3), it allowed VRE to colonize the gut to very high levels in most of the mice (Fig. 1d, Supplementary Fig. 4), indicating that a milder degree of dysbiosis can cause a similar impact on colonization resistance. Despite high levels of VRE intestinal colonization, no signs of pain, distress or discomfort were detected in mice after VRE inoculation. Next, we evaluated the capacity of the microbiota to recover after antibiotic cessation and how this recovery would affect colonization resistance against VRE. To this end, we repeated the previous experiment except that we allowed the microbiota to recover for two weeks before analysing its composition and challenging the mice with VRE. Confirming previous results by our and other groups[7,10,20], antibiotic-induced dysbiosis persisted 2 weeks after antibiotic cessation (Fig. 1). Nevertheless, certain degree of microbiota recovery (i.e. biomass, richness, diversity,

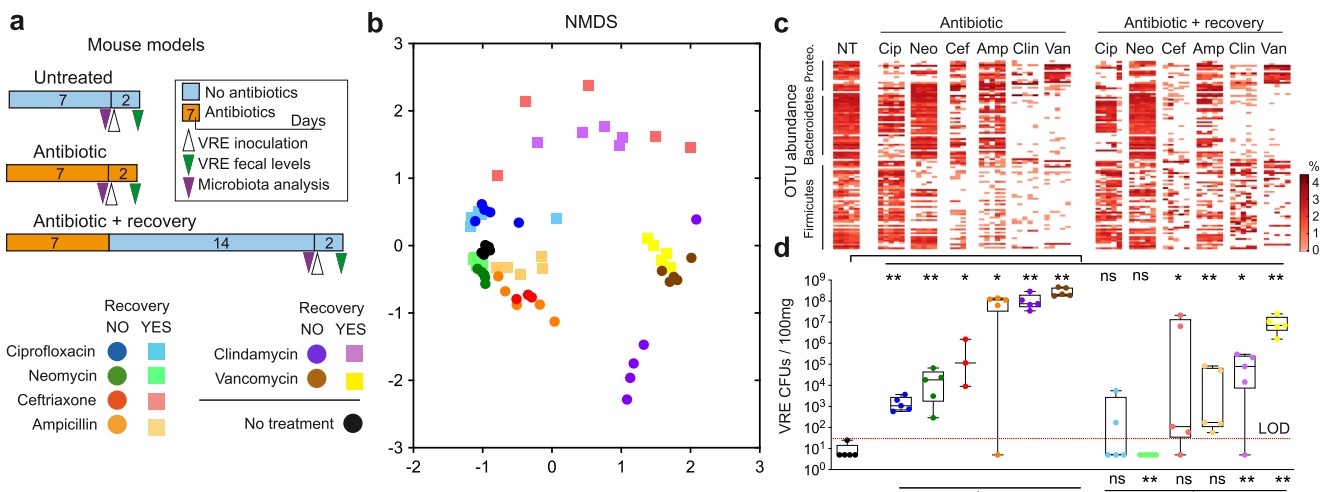

**Fig. 1 | Different antibiotics induce distinct dysbiotic states and grades of susceptibility to vancomycin-resistant *Enterococcus* (VRE) intestinal colonization. a** Schematic representation of the mouse model. Mice were treated during seven days with antibiotics of different spectrum (i.e. ciprofloxacin, neomycin, ceftriaxone, ampicillin, clindamycin or vancomycin). Subsequently, a group of mice was orally gavage with $10^6$ VRE colony forming units (CFUs), while another group of mice was allowed to recover for two weeks before VRE inoculation. Faecal samples were collected immediately before VRE inoculation for microbiota analysis and 2 days post-VRE inoculation (p.i.) for quantifying VRE levels. As control, faecal samples were collected from a group of untreated mice for microbiota analysis and VRE quantification. **b** Non-metric multidimensional scaling (NMDS) analysis based on Bray-Curtis distances obtained using the relative abundance of OTUs identified in faecal samples collected from mice. Each point represents the microbiota of one mouse. Color legend is shown in (**a**) and indicates the antibiotic treatment received. **c** Heatmap that shows the abundance of the top 100 most abundant OTUs identified in the faecal samples collected. No treatment (NT), Ciprofloxacin (Cip), Neomycin (Neo), Ceftriaxone (Cef), Ampicillin (Amp), Clindamycin (Clin), Vancomycin (Van), Proteo (Proteobacteria). Taxonomy of the OTUs as well as statistical analysis of the OTUs abundance are indicated in Supplementary Data File 2. **d** VRE faecal levels (CFUs / 100 mg) 2 days p.i. in the mice receiving the different treatments. LOD = limit of detection. Points below the LOD indicate those mice in which we were not able to detect any VRE CFU. *$p < 0.05$, **$p < 0.01$, ns – nonsignificant, two-sided Wilcoxon rank-sum test. $N = 5$ mice per treatment except for ceftriaxone without recovery period in which $N = 3$ mice. Statistical results shown in panel (**d**) refer to: above - the comparison of each treated group of mice with the untreated group; below: the comparison of each group of mice treated with a specific antibiotic (before vs after the recovery period). In (**d**) boxes extend from the 25th to 75th percentiles. The line within the boxes represents the median. Whiskers indicate the maximum and minimum values. Source data are provided as a Source Data file.

taxa and OTUs) was detected for most antibiotics (Fig. 1, Supplementary Fig. 2, Supplementary Fig. 3). Consistent with a partial recovery of the microbiota, the capacity of VRE to colonize the intestinal tract was in most cases reduced 2 weeks after antibiotic cessation (Fig. 1d, Supplementary Fig. 4).

## A consortium of five commensal bacteria restricts vancomycin-resistant *Enterococcus* intestinal colonization during antibiotic-induced dysbiosis

The previous results indicate that antibiotics promote distinct changes in the microbiota, which allow VRE to colonize the intestine to different levels. Subsequently, we compared the specific microbial changes induced by each antibiotic in each mouse with the capacity of VRE to colonize the intestinal tract. This analysis aims to detect specific microbial populations that potentially confer resistance in untreated mice and whose depletion by antibiotic treatment promotes VRE intestinal colonization. Spearman correlation analysis identified specific genera whose abundance was negatively associated with VRE colonization (Supplementary Fig. 5; Supplementary Data File 3). To verify the potential inhibitory effect of the identified commensal bacteria on VRE gut colonization, we performed extensive culturing of caecal contents from untreated mice (see methods). Notably, we were able to isolate bacterial strains from multiple abundant genera (median abundance in untreated mice >0.4%) that were negatively associated with VRE intestinal colonization (Spearman, rho <−0.31; q < 0.05; Supplementary Fig. 5). Specifically, strains from the genera *Alistipes, Barnesiella, Olsenella and Oscillibacter* were isolated. In addition, we obtained an isolate defined as *Ruminococcaceae*_Unclassified (*Ruminococcaceae_UC*), a prevalent bacterial group (median relative abundance in untreated mice = 6.37%) that could not be classified to the genus level based on its 16S rRNA sequence, but that was highly significantly associated with

resistance against VRE (Spearman rho = −0.64, q=1.7e-6, Supplementary Data File 3). Consistent with a role of these taxa in restricting VRE gut colonization, mice highly susceptible to VRE intestinal colonization contained lower levels of these specific taxa as compared to more resistant mice (Fig. 2a). To confirm the role of this commensal bacterial consortium (CBC) in limiting VRE intestinal colonization, we tested if its inoculation to mice that had been treated with vancomycin would diminish VRE gut levels (Fig. 2b). We chose vancomycin for this test because: (i) it is the antibiotic that allows the highest level of VRE colonization after antibiotic cessation despite the recovery of the total biomass (Fig. 1d; Supplementary Fig. 2C), (ii) it promotes permanent changes in the microbiota (Fig. 1b), including the permanent depletion of the 5 isolated taxa included in CBC (Supplementary Fig. 6) and (iii) it is frequently used in the clinic and induces similar microbiota changes in patients and mice, promoting in both cases VRE intestinal colonization[10,21]. As shown in Fig. 2c, 2 weeks after vancomycin cessation, mice that had received CBC were significantly more resistant to VRE colonization than those mice that received phosphate buffer saline - glycerol 20% - cysteine 0.1% (PBS-GC; the vehicle for bacteria preservation). Seven days post-VRE inoculation, the number of VRE CFUs detected in faeces was >3 orders of magnitude lower in those mice that received CBC as compared to those mice that received PBS-GC. This capacity of restricting VRE gut colonization was not detected with the administration of another bacterium: *Bacteroides*, a highly abundant taxa (average= 12.38%, including all samples from Fig. 1) that was not associated with protection against VRE levels but that was also permanently depleted after vancomycin treatment (Supplementary Data File 3, Supplementary Fig. 7). Notably, a similar effect of CBC on gut colonization by resistant enterococci was detected for additional enterococcal strains tested (AUS0004, E1162, Supplementary Fig. 8). The CBC effect of restricting VRE colonization was associated with the

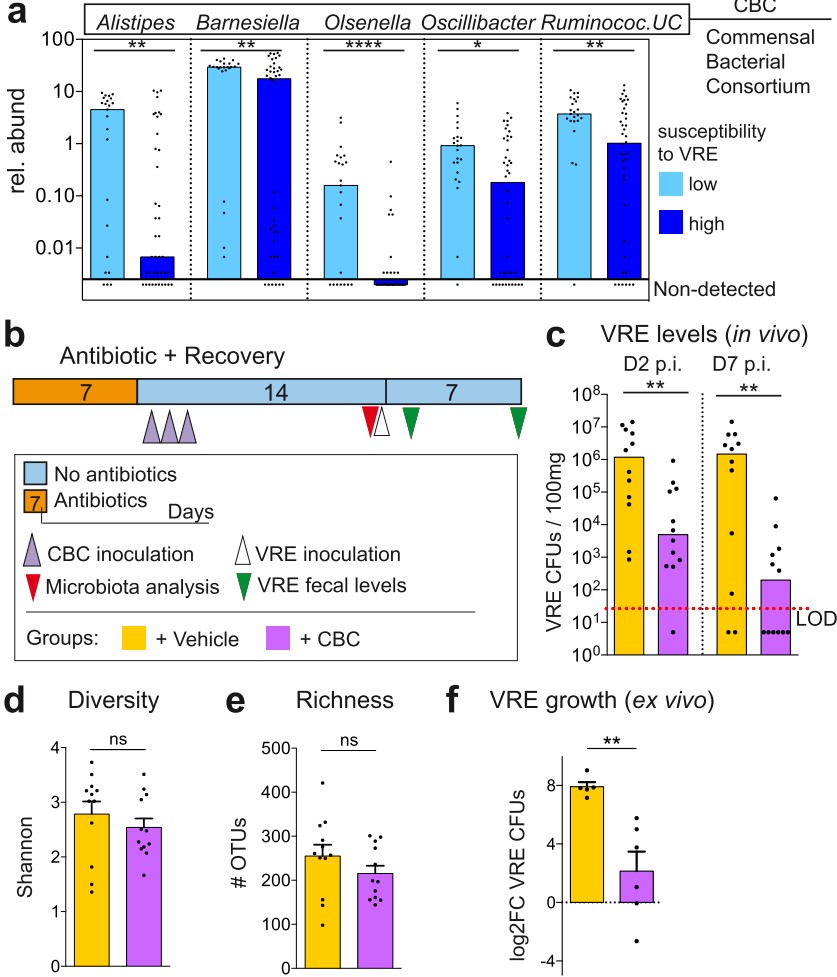

**Fig. 2 | A consortium of five commensal bacteria restricts vancomycin-resistant *Enterococcus* (VRE) intestinal colonization during antibiotic-induced dysbiosis. a** Faecal relative abundance (rel. abund) of the five bacterial taxa whose levels were negatively associated with VRE faecal levels (Supplementary Fig. 5, Supplementary Data File 3) and that could be isolated, UC = unclassified. Samples (from Fig. 1) are divided in those collected from mice with high ($> 10^3$ VRE colony forming units (CFUs) / 100 mg) or low ($< 10^3$ VRE CFUs / 100 mg) susceptibility to VRE intestinal colonization (see methods for the definition of the two groups). CBC (Commensal Bacterial Consortium) is the acronym given to the bacterial mix containing the 5 isolates from the 5 taxa indicated. ****$p = 4e^{-5}$,**$p < 0.01$,*$p = 0.011$. **b** Schematic representation of the mouse model used. Mouse were treated with vancomycin. One day after antibiotic cessation, mice received CBC orally during 3 consecutive days. Two weeks after antibiotic cessation, a faecal sample was collected immediately before VRE inoculation. VRE faecal levels were quantified 2 and 7 days post-VRE inoculation (p.i.). As control, a group of mice received the vehicle for bacteria administration (PBS-GC) instead of CBC. **c** VRE levels in faecal samples collected from the group of mice described in (**b**), **$p < 0.006$. **d** Microbiota diversity and (**e**) microbiota richness (number of Operational Taxonomical Units – OTUs) from the groups of mice described in (**b**), ns$p > 0.21$. **f** Ex vivo VRE growth in caecal contents collected from vancomycin-treated mice that received CBC or PBS-GC after stopping antibiotic treatment. Caecal contents were collected 2 weeks after stopping antibiotic treatment. Each point is a biological replicate (i.e. caecum collected from a different mouse). Values represent the change (log2FC) in VRE levels after 24 h of growth, **$p = 0.004$. Two-sided Wilcoxon rank-sum test for (**a** and **c**) and two-sided *t*-test for (**d**–**f**), ns - nonsignificant. $N = 23$ and 40 mice per group in (**a**), $N = 12$ mice per group in (**c**–**e**), $N = 5$ biologically independent samples per group in (**f**). Bars represent the median in (**a** and **c**) and the mean in (**d**–**f**). Whiskers represent the SEM. Source data are provided as a Source Data file.

restoration of the levels of the bacterial isolates administered (Supplementary Fig. 9) but not with an overall increase in microbiota richness or diversity (Fig. 2d, e). Unexpectedly, mice that did not received CBC, spontaneously recovered bacteria from the taxa *Oscillibacter and Ruminococcaceae_UC* (Supplementary Fig. 9), indicating that the recovery of these taxa was not sufficient to limit VRE gut colonization. Importantly, CBC inhibitory effect against VRE was also detected ex vivo (Fig. 2f). VRE was able to grow in intestinal contents collected from mice that recovered from vancomycin treatment. In contrast, VRE growth was significantly reduced in the intestinal contents collected from mice that had received CBC after vancomycin administration. This last result suggests that CBC can directly inhibit VRE growth in the absence of host-derived cells and prompt us to investigate bacterial functions restored by CBC administration that could be involved in VRE inhibition.

## A commensal bacterial consortium restores the expression of specific bacterial functions that were lost after vancomycin treatment, including functions related to the metabolism of fructose

Next, we attempted to decipher mechanisms by which CBC restricts VRE gut colonization by comparing the bacterial functions expressed in the microbiomes of untreated mice or mice treated with vancomycin that received either CBC or the bacterial vehicle PBS-GC after antibiotic cessation. In addition, we sequenced the genomes of the CBC isolates, which allow us to identify those bacterial functions specifically encoded and expressed by these bacteria. Moreover, the complete genome of the sequenced isolates allowed us to classify to the genus level (*Flavonifractor*; Supplementary Fig. 10) the isolate previously defined as UC_Ruminococcaceae. We first evaluated that besides gut colonization (Supplementary Fig. 9), the administered

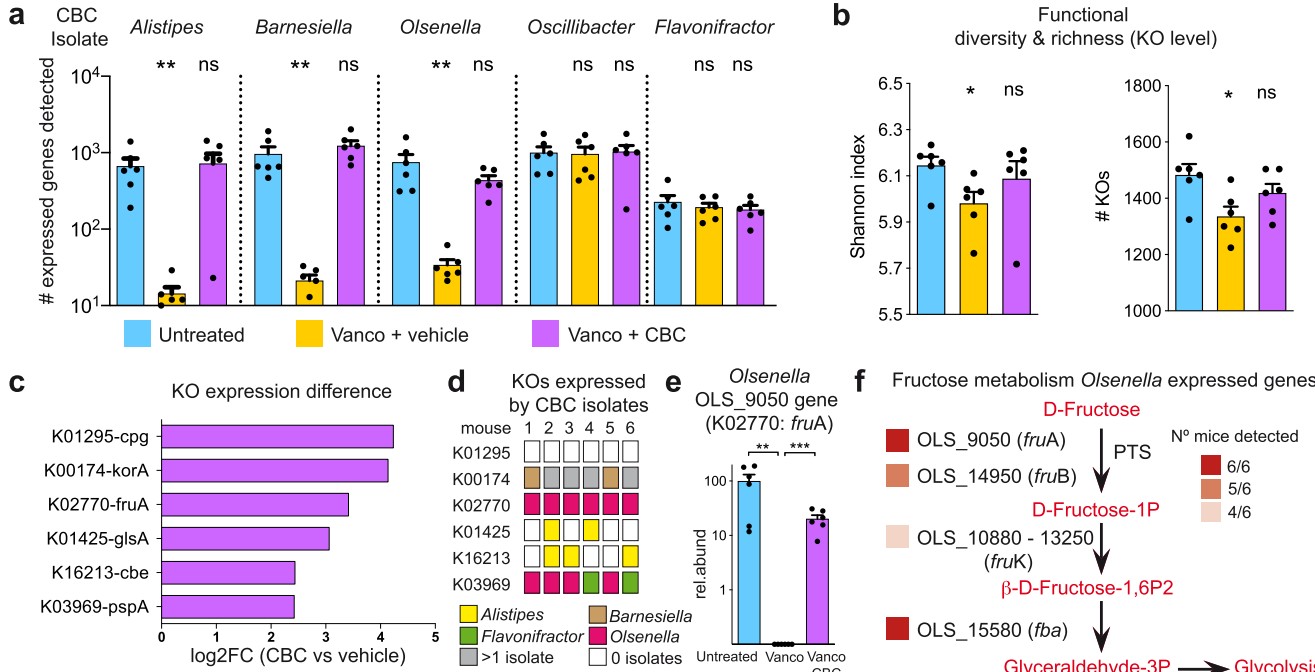

**Fig. 3 | A commensal bacterial consortium (CBC) restores the expression of specific bacterial functions that were lost after vancomycin treatment, including functions related to the metabolism of fructose. a** Number of genes encoded by CBC isolates whose expression was detected in the caecum contents of untreated mice and mice that were treated with vancomycin and received either PBS-GC (Vanco+vehicle) or CBC (Vanco+CBC), $**p < 0.0054$, $^{ns}p > 0.169$. **b** Diversity and richness of the KEGG orthologs (KOs) expressed in the 3 groups of mice described in (**a**), $*p < 0.018$, $^{ns}p > 0.228$. **c** KOs whose expression is significantly higher ($>2\log_2$ fold change (FC); $q < 0.05$) in the group of mice Vanco+CBC as compared to the group Vanco+vehicle and whose abundance does not differ between Vanco+CBC and Untreated. An extended list of KOs whose expression was increased upon CBC administration ($>1\log_2$FC) is shown in Supplementary Data File 8. **d** Color panel that indicates if any of the isolates of CBC express the KOs indicated in (**c**) in each one of the mice from the group Vanco+CBC. **e** Expression in

the different mouse groups of the gene encoded by *Olsenella* that was assigned to K02770. This gene encodes a subunit of a PTS transporter for fructose (FruA). The level of expression (rel.abund) for each mouse is calculated as the % of the expression detected in one mouse as compared to the average expression detected in untreated mice, $**p = 0.0097$, $***p = 0.0002$. **f** Expression of genes encoded by *Olsenella* that are required for the internalization and metabolization of fructose. The heatmap indicates the number of mice (red colors) in which the expression of that particular gene was detected in the group (Vanco+CBC). In the case of *fruK*, it was found to be expressed by one gene copy in some mice and another gene copy in others, so both genes are indicated. Two-sided t-test (**a**, **b** and **e**), ns - non-significant. Statistics in (**a** and **b**) refer to the comparison with the untreated group. $N = 6$ mice per group. Bars in (**a**, **b** and **e**) represent the mean. Whiskers represent the SEM. Source data are provided as a Source Data file.

bacteria were transcriptionally active. The expression of specific genes encoded by *Barnesiella*, *Alistipes* and *Olsenella* isolates could be detected in vancomycin-treated mice that received CBC as well as in untreated mice (Fig. 3a). However, expression for the majority of these genes was undetectable in vancomycin-treated mice that did not receive CBC. This result indicates that these three isolates were able to colonize the gut and restore the expression of specific bacterial genes that were lost upon antibiotic treatment. In contrast, expression of genes encoded by the *Oscillibacter* and *Flavonifractor* isolates were identified in all three groups of mice. This is consistent with the spontaneous recovery of bacteria from these taxa (Supplementary Fig. 9) and strongly suggests that these two taxa are not sufficient to prevent VRE gut colonization. Next, we evaluated to what extent the administration of CBC could modify the functional capacity of the microbiome. To this end, KEGG orthologs (KO) were assigned to non-redundant genes identified in the murine caecal transcriptome (see methods). Diversity and richness of the expressed KOs were significantly decrease after vancomycin treatment and partially recovered by CBC administration (Fig. 3b). Subsequently, we defined those specific bacterial functions (KOs) whose expression was enhanced after CBC administration to vancomycin treated mice and that therefore could be relevant for limiting VRE gut colonization. To this end, DeSeq2 statistical analysis was applied to identify KOs whose expression was significantly increased ($\log_2FC > 2$; $q < 0.05$) upon CBC administration and that reach similar levels as the expression detected in untreated mice. A total of 6 KOs matches the defined criteria

(Fig. 3c). Most of these KOs were assigned to enzymes which have not been described to contribute to colonization resistance against pathogens, including K01295 (peptidase), K00174 (oxidoreductase), K01425 (glutaminase). In addition, K03969, putatively encoding a phage shock protein, was also overexpressed in the group of mice that received CBC. Enterococcal phages have been shown to diminish intestinal colonization by a clinical *Enterococcus* isolate[22]. We found in most of the CBC-inoculated mice that one of the introduced isolates (*Olsenella*) expressed the K03969 (Fig. 3d). *Olsenella* is an Actino-bacteria and therefore phylogenetically unrelated to *Enterococcus* (Firmicutes). Taking into account the very narrow spectrum of phages, we assumed that it was unlikely that a phage present in *Olsenella*, encoding this KO, could be contributing to resistance to VRE colonization. Noteworthy, two additional KOs (K16213 and K02770) that were overexpressed in the group of mice that received CBC, encoded for proteins required for the uptake and catabolism of specific sugars. Theoretically, the expression of these KOs could contribute to VRE inhibition through sugar depletion, a colonization resistant mechanism that has been well characterized in *Enterobacteriaceae* species but not in VRE. K16213 encodes a cellobiose epimerase while K02770 encodes for FruA, a subunit of a PTS specific for the transport of fructose. The K16213 KO was found to be expressed by CBC isolates in only 50% of the analysed mice (Fig. 3d). However, expression of the KO encoding for the fructose transporter was detected in one of the administered isolates (i.e. *Olsenella*) in all analysed mice (Fig. 3d). The genome of the isolated *Olsenella* contains 4 genes that encode for FruA

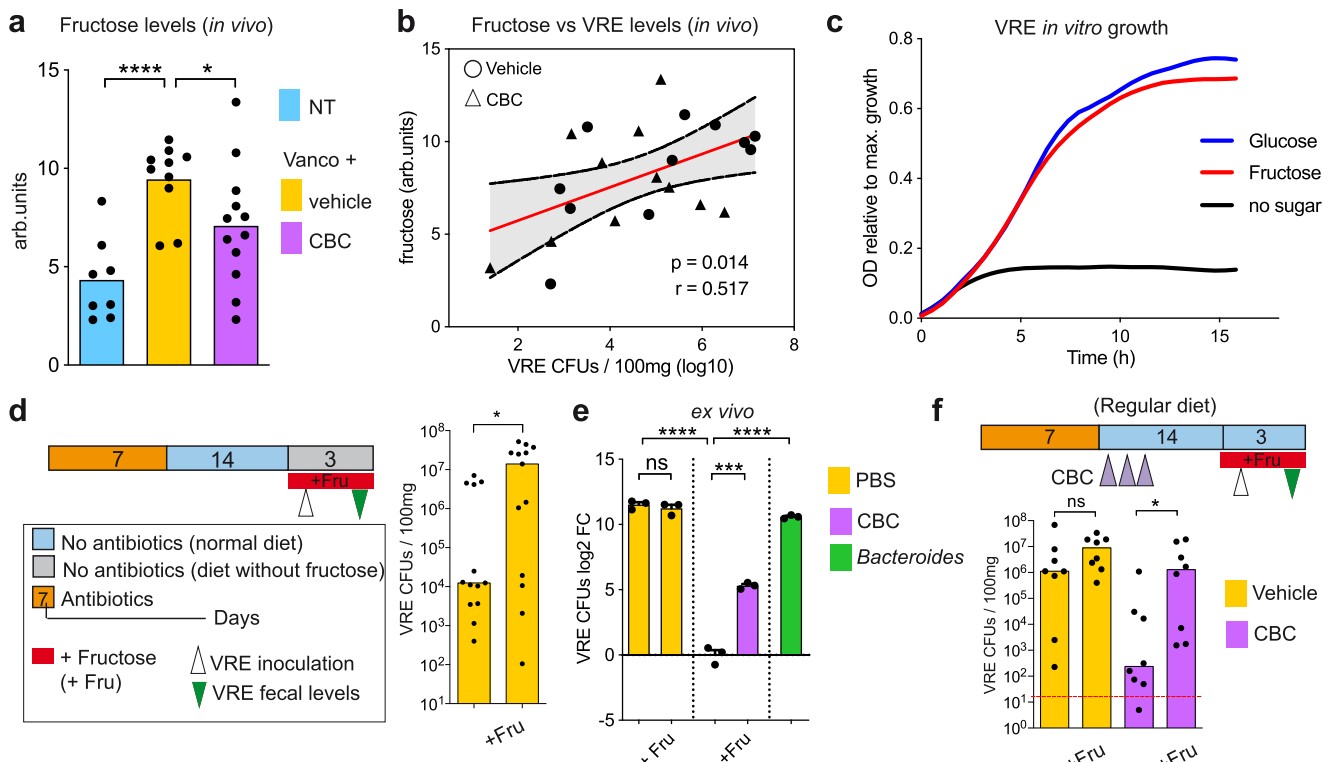

**Fig. 4 | The commensal bacterial consortium (CBC) restricts vancomycin-resistant *Enterococcus* (VRE) intestinal colonization through nutrient competition by depleting fructose, a sugar that boosts VRE growth in vivo. a** Caecal fructose levels of mouse groups described in Fig. 3. arb.units: arbitrary units,****$p = 5.1e\text{-}5$,*$p = 0.047$. **b** Two-sided Pearson correlation analysis between the caecal fructose levels and faecal $\log_{10}$ VRE colony forming units (CFUs) detected in paired co-housed mice 2 days after VRE inoculation, $p = 0.014$. The red line represents the linear regression mean and the dotted line the 95% CI. **c** VRE growth in minimal medium containing fructose, glucose or no extra carbon source. The figure indicates the relative growth compared to the sugar that allowed a highest growth (i.e. maltotriose, Supplementary Fig. 13). **d** Mice were inoculated with VRE 2 weeks after vancomycin treatment cessation. One day before, mice were fed with a diet that do not contain fructose (see methods) and half of the mice received fructose (Fru) in their drinking water. VRE levels were quantified in faeces 2 days after VRE inoculation, *$p = 0.043$. **e** Filtered caecal contents from mice that recovered from

vancomycin treatment were inoculated with PBS, CBC or *Bacteroides*. 24 h after, VRE was inoculated and the change in VRE levels was quantified 6 h later. + Fru indicates that immediately before VRE inoculation an excess of fructose was added to the culture,ns$p = 0.459$,***$p = 0.0002$,****$p < 1.2e\text{-}5$. **f** 2 weeks after vancomycin cessation mice were inoculated with VRE. One day after vancomycin cessation mice received CBC or the bacterial vehicle (PBS-GC) instead. Half mice from each group received fructose in the drinking water, starting one day before VRE inoculation. Faecal VRE levels were quantified 2 days post-inoculation, ns$p = 0.16$, *$p = 0.021$. Two-sided t-test for (**a** and **e**); two-sided Wilcoxon rank-sum test for (**d** and **f**), ns – nonsignificant. $N = 8$, 10 and 12 mice per group in (**a**). $N = 22$ mice in (**b**). $N = 13$ mice per group in (**d**). $N = 3$ biologically-independent samples per group in (**e**). $N = 8$ mice per group in (**f**). Bars represent the mean in (**a** and **e**) and the median in (**d** and **f**). Red line in (**f**) represents the limit of detection. Source data are provided as a Source Data file.

(KO2770). However, expression of FruA by *Olsenella* was mainly driven by one single gene: OLS_9050, in all the six analysed mice (Fig. 3e). As expected, OLS_9050 expression was not detected in vancomycin-treated mice not receiving CBC but was detected in all untreated mice (Fig. 3e). In addition to FruA, in most of the analysed mice that received CBC (66–100%), it was possible to detect transcripts expressed by the administered *Olsenella* isolate that encoded for the other subunit of the fructose transporter (FruB:K02768) as well as the other genes required for the incorporation of fructose into the glycolytic pathway: *fru*K (K00882) and fructose-biphosphate-aldolase (*fba*, K01624) (Fig. 3f). These results suggested that CBC and more specifically *Olsenella* express in vivo bacterial functions for the uptake and utilization of fructose. This was further supported by in vitro experiments showing that *Olsenella* can grow in vitro using fructose as carbon source (Supplementary Fig. 11).

### The commensal bacterial consortium restricts intestinal VRE growth through fructose depletion

Fructose can be found in the large intestine[23,24], and previous studies have shown that some enterococcal isolates can use this sugar as a nutrient source[25,26]. Therefore, taking into account the transcriptomic

data, CBC could be restricting VRE gut colonization through fructose depletion. To test this hypothesis, we first check the levels of fructose in the caecal content of the mice in which we had analysed their transcriptome and in additional mice that followed a similar treatment (i.e. additional untreated mice and mice that recovered from vancomycin treatment that received either CBC or PBS-GC, Fig. 4a). Fructose levels were significantly higher in those mice that received vancomycin as compare to untreated mice (Fig. 4a). As expected, administration of CBC significantly reduced the caecal levels of fructose (Fig. 4a). To evaluate how caecal fructose levels impact the VRE capacity to colonize the intestinal tract, we first performed a correlation analysis between the caecal levels of fructose and the VRE faecal levels of paired co-housed mice that were orally inoculated with VRE. Note that mice that have been co-housed in the same cage are equally susceptible to VRE intestinal colonization (Supplementary Fig. 12). We used this indirect approach since (i) the sugar content of the caecum likely better represents the amount of sugars available for bacterial growth than the levels of sugar excreted in faeces and (ii) analysing the caecal levels of fructose post-VRE inoculation and the VRE colonization levels in the same mouse could be misleading since VRE could have an impact on fructose levels. Notably, the fructose levels of the caecum

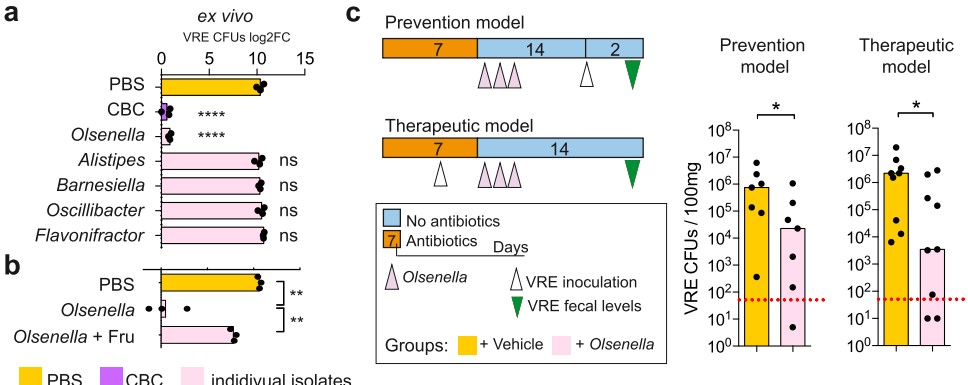

**Fig. 5 | *Olsenella* recapitulates the inhibitory effect of the bacterial consortium (CBC) against vancomycin-resistant *Enterococcus* (VRE) colonization.**
**a**, **b** Filtered caecal contents from mice that recovered from vancomycin treatment were inoculated with either PBS, CBC or individual isolates as depicted. 24 h after incubation at 37 °C under anaerobic conditions, VRE was inoculated and the growth was quantified 6 h later. Values represent the change (log2) in VRE levels after the 6 h. + Fru indicates that immediately before VRE inoculation an excess of fructose was added to the culture, ****$p < 1.17e-5$,**$p < 0.005$,$^{ns}p > 0.188$. **c** Prevention model: mice were treated with vancomycin and one day after stopping antibiotic treatment received either *Olsenella* or PBS-GC for 3 consecutive days. Two weeks after antibiotic withdrawal mice were inoculated through oral gavage with VRE and VRE

colony forming units (CFUs) were quantified 2 days later in faeces. Therapeutic model: mice were treated with vancomycin for 4 days and inoculated with VRE. Mice were maintained on vancomycin for 3 more days. One day after stopping antibiotic treatment, mice received *Olsenella* for 3 consecutive days or PBS-GC. Two weeks after stopping antibiotic treatment VRE levels were measured in faeces, *$p < 0.05$. Two-sided $t$-test for (**a** and **b**). Statistics in (**a**) refer to the comparison with the PBS group, ns – nonsignificant. One-sided Wilcoxon rank-sum test for (**c**). $N = 3$ biologically independent samples in (**a** and **b**). $N = 7$ mice per group in the prevention model and $N = 9$ mice per group in the therapeutic model in (**c**). Bars represent the mean in (**a** and **b**) and the median in (**c**). Red line in (**c**) represents the limit of detection. Source data are provided as a Source Data file.

significantly and positively correlated with the VRE faecal levels of paired co-housed mice (Fig. 4b). This result suggests that fructose promotes the expansion of VRE in vivo and its depletion by CBC impairs VRE intestinal colonization. To confirm this hypothesis, we first checked that the VRE strain used in this study can efficiently use fructose as a carbon source. In vitro experiments using an array of carbon sources confirmed that fructose represents one of the carbohydrates that promotes higher growth of VRE under anaerobic conditions (i.e., Supplementary Fig. 13). Indeed, VRE was able to grow in fructose to a similar extent as in glucose, one of the central sugars of bacterial metabolism (Fig. 4c). Next, to further characterize the role of fructose in VRE gut colonization, we pursued a mutagenesis approach in order to obtain a VRE strain not capable of utilizing fructose. To this end, we aimed to delete from the VRE genome all the fructokinases that are involved in the first step of fructose metabolism. We identified 6 genes encoding potential fructokinases (including also 1-phospho-fructokinases) in the VRE strain used in the majority of the experiments (ATCC70021). In addition, 5 fructokinases were found to be encoded by the other VRE strain used in this work (Supplementary Fig. 8, AUS0004). This result suggests a high level of functional redundancy for enzymes required for fructose utilization in VRE, which may be indicative of a key role for fructose in VRE growth. Subsequently, we attempted to obtain a VRE mutant lacking all potential fructokinases in the genetically-tractable strain AUS0004[27]. However, after multiple attempts (see methods), we were not able to disrupt 2 out of the 5 genes encoding potential fructokinases, indicating that, at least in the tested conditions, two of the fructokinases may be key for VRE growth. Nonetheless, we found that the growth of the 3 mutants that could be generated was reduced in the presence of fructose but not in the presence of other sugars frequently found in the gut (i.e., glucose, mannose, Supplementary Fig. 14), confirming that multiple genes in the VRE genome encode for functional fructokinases. This genetic functional redundancy further supports a key role for fructose in VRE metabolism, however, it precludes studying the effect of fructose on VRE gut colonization using a mutagenesis approach without having a proper strain lacking all fructokinases. For this reason, we decided to use an alternative strategy to investigate the relevance of fructose in VRE intestinal colonization. To this end, we fed mice with a customized

diet that does not contain fructose, or the same diet but supplementing fructose in the drinking water (see methods), expecting that the presence of fructose would give a growth advantage to VRE in the gut environment. Notably, VRE reached significantly higher levels (i.e., 3 orders of magnitude) in those mice that received fructose in their drinking water as compared to mice not receiving fructose (Fig. 4d). This result demonstrates that fructose boosts VRE growth in the intestinal tract.

Previous studies have identified commensal bacteria that can suppress VRE intestinal colonization through production of inhibitory molecules. However, our results suggested that CBC was inhibiting VRE growth through nutrient competition (i.e. fructose depletion). To further confirm this hypothesis, we made use of an ex vivo assay. Briefly, the caecal contents of mice that recovered from vancomycin treatment were filtered and reduced in an anaerobic chamber (see methods). The filtered contents were then incubated in the presence of CBC or a *Bacteroides* isolate (a non-protective commensal) to allow nutrient depletion. Subsequently, VRE was inoculated and its growth was monitored 6 h after. VRE growth could be detected in filtered intestinal contents that did not contained any bacteria or that had been incubated with *Bacteroides* (Fig. 4e). In contrast, previous incubation with CBC completely abolished VRE growth (Fig. 4e). To demonstrate that CBC was suppressing VRE growth through depletion of key nutrients (i.e. fructose) and not through production of inhibitory molecules, an excess of fructose, so that fructose would not become a limiting nutrient source, was added to the filtered caecal contents after being incubated with CBC and immediately before VRE inoculation. Notably, addition of an excess of fructose was sufficient to restore VRE growth in the presence of CBC (Fig. 4e). Importantly, this result could also be reproduced using the in vivo mouse model. Administration of an excess of fructose in the drinking water to mice that were allowed to recover from vancomycin treatment and were fed with a regular chow diet (see methods) restored the capacity of VRE to colonize the large intestinal tract in the presence of CBC (Fig. 4f). Altogether these results suggest that CBC suppresses VRE growth through depletion of key nutrient sources for VRE growth, in particular, through depletion of fructose, a sugar that promotes the expansion of VRE in the intestinal tract.

### *Olsenella* can restrict VRE intestinal colonization

Results from the transcriptomic data suggested that *Olsenella* was the bacterium from the CBC consortium that was responsible for fructose depletion. Therefore, we decided to investigate if this bacterium was key in restricting VRE intestinal colonization. To this end, we first repeated the ex vivo experiment but incubating the intestinal contents with each bacterium of the CBC consortium separately. None of the bacteria from the CBC consortium except *Olsenella* was capable of suppressing VRE growth (Fig. 5a), despite similar levels of each commensal bacterium were detected previous to VRE inoculation (Supplementary Fig. 15). As with CBC, VRE growth in the presence of *Olsenella* could be restored by adding an excess of fructose immediately before VRE inoculation (Fig. 5b). This result suggested that *Olsenella* was the key bacterium responsible for restricting VRE gut colonization. This was further confirmed in vivo. Mice treated with vancomycin, that received *Olsenella* after antibiotic cessation, were significantly less susceptible to VRE intestinal colonization than mice that received only PBS-GC upon antibiotic withdrawal (Fig. 5c, prevention model). In addition, the administration of *Olsenella* to mice that were previously colonized with VRE (Supplementary Fig. 16), allowed a significant reduction of VRE levels (>2 orders of magnitude) as compared to mice that received PBS-GC instead (Fig. 5c, therapeutic model). This last result indicates that besides preventing VRE intestinal colonization, *Olsenella* was capable of displacing VRE from the intestinal tract. However, since genetic manipulation of *Olsenella* species has not been implemented yet, the role of specific *Olsenella*-encoded genes (e.g. *fru*A) in restricting VRE colonization could not be evaluated in this study.

## Discussion

Intestinal colonization by multidrug resistant pathogens, including VRE, can represent the first step towards the development of infections that have very limited or no treatment options. Finding approaches to limit gut colonization by resistant pathogens could restrict their spread to other patients by limiting the pathogen shedding in faeces on top of preventing systemic infections[9,11]. Using a combination of omic techniques, in vitro, ex vivo and in vivo experiments, we have identified a commensal bacterium (i.e. *Olsenella*) that can restrict VRE gut colonization. Moreover, we have identified a novel mechanism by which commensal bacteria can prevent VRE gut colonization: diminishing the availability of nutrients, specifically of fructose, a sugar that boosts VRE growth.

Nutrient requirements for gut colonization by pathogens of the family *Enterobacteriaceae* has been extensively studied[16,28–30]. However, the nutrient sources required for *Enterococcus* gut colonization are largely unknown. A recent study shed some light into this question and identified lactose as a sugar that promotes *Enterococcus* growth in the murine intestine[18]. We attempted to quantify the levels of lactose in the caecal content from mice to investigate if CBC could also be contributing to VRE inhibition through lactose depletion. However, we were not able to detect lactose in the caecal contents of the mice from our study (Supplementary Fig. 17). This result suggests that either lactose is depleted by commensals that recovered after vancomycin treatment (independent of CBC administration) or that most lactose was digested and absorbed by epithelial cells in the small intestine. Thus, in our mouse model, lactose depletion does not seem to be involved in the CBC mediated inhibition of VRE growth in the large intestine. In contrast, we found detectable levels of fructose in the caecum of mice, which correlated with the capacity of VRE to colonize the gut. Moreover, fructose boosted VRE growth in vivo and in vitro suggesting a key role for this sugar in the VRE colonization process. Consistent with the relevance of fructose as a nutrient source, in a recent study it was described that all the 24 *Enterococcus* analyzed species, spanning the diversity from this genus, can use fructose to grow and their core genome encodes genes required for fructose utilization[31]. More specifically, the strain used in the majority of

experiments in our study (ATCC700221) encodes for multiple copies of genes required for the metabolism of fructose[32], including 6 copies codifying for the initial enzymes (fructokinases) required for utilizing fructose once it has been internalized (similar to *fru*K in Fig. 3). In addition, the other two resistant enterococci strains used in our study (AUS0004 and E1162) contain 5 potential fructokinases. This level of genetic redundancy also supports that this nutrient source is relevant for VRE growth but hinders its functional study through genome mutagenesis. In fact, we attempted to delete all 5 fructokinases (including also 1-phosphofructokinases) in the VRE genetically-tractable strain AUS0004[27]. However, after several attempts, including the implementation of a novel methodology that facilitates mutagenesis through overexpression of a recombinase[33], we were not able to obtain mutants for two of the potential fructokinases, which preclude our ability to completely abolish fructose utilization by VRE and study its impact on gut colonization using a mutagenesis approach.

Diet and mucins (the major component of the intestinal mucus) represent two of the major sources of sugars consumed by intestinal bacteria[34,35]. Fructose is not frequently incorporated into mucins[35], but it is widely found in the diet as free monosaccharides, the disaccharide sucrose and as polymers (fructans)[34]. One of the major components of the mouse diet used in our study is wheat, which contains free fructose and fructans[36]. Although free fructose can be absorbed in the small intestine[37], the majority of fructans are not absorbed[38] and reach the large intestine. There, fructans can be hydrolysed into fructose by extracellular bacterial enzymes (fructanases, including beta-fructosidases and levanases) and the liberated fructose is then internalized through specific transporters to serve as a nutrient source[39]. Therefore, commensals encoding fructanases could be promoting VRE colonization. In line with this hypothesis, preliminary analysis of the transcriptomic data detected a positive association, albeit not statistically significant, between the expression levels of beta-fructosidases (a type of fructanase) and the faecal VRE levels (Supplementary Fig. 18). Interestingly, this association was detected in caecal samples from vancomycin-treated mice (Supplementary Fig. 18A; $r = 0.76$, $p = 0.079$), but it was lost in samples from vancomycin-treated mice that had received CBC (Supplementary Fig. 18B; $r = -0.02$; $p = 0.95$). Since the expression levels of beta-fructosidases did not differ between both groups of mice (Two-sided Wilcoxon rank-sum test, $p > 0.99$), this lack of association in mice that received CBC might be explained by the consumption of available fructose by the bacterial consortium. Nevertheless, we acknowledge that this analysis was performed with a limited number of mice ($N = 6$) and that further studies should be performed in order to validate this hypothesis and identify potential commensal bacteria from the microbiome that may enhance VRE intestinal colonization through liberation of nutrients, including fructose. In addition to fructans, free fructose that has not been absorbed in the small intestine could be a carbon source for bacterial growth in the large intestine. In humans, the capacity of the small intestine to uptake fructose is limited and is frequently exceeded due to the consumption of foods or beverage rich in fructose. For example, soft drinks that contain high fructose corn syrups as sweeteners and that have been introduced in the consumer habits of modern societies[34]. The excess of fructose reaches the large intestine which leads to several symptoms, such as bloating and abdominal pain, derived from its fermentation by commensal bacteria[40]. Our study pinpoints an additional potential negative effect of high-fructose intake: increasing the gut levels of multidrug resistant pathogens such as VRE that can expand using this sugar as a carbon source.

Since intestinal colonization is key for the development of subsequent infections in hospitalized patients[6], novel approaches, based on the microbiome, have been proposed to restrict VRE gut colonization. One of such strategies is the restoration of the microbiota of patients by administering the faecal microbiota of a healthy donor (i.e. faecal microbiota transplant - FMT)[41]. However, such approach could

be problematic due to the incompletely defined composition of the FMT and the subsequent risk of introducing potentially pathogenic bacteria or virus[42]. This will be even more risky in VRE infected patients, which are usually immunocompromised. For this reason, alternative strategies, involving the introduction of defined bacterial consortia or single strains have been proposed[8,12]. In this study, we have identified *Olsenella sp* as a novel candidate that could be used for restricting VRE gut colonization. Although our isolate has a murine origin, several species from the same genus have been isolated from humans, both in the oral cavity and faeces[43,44]. Interestingly, one of these species (*Olsenella profusa*) is able to ferment multiple intestinal carbohydrates, including fructose. Future studies that test the impact of this human isolate on VRE intestinal colonization are warranted. In addition, as an alternative, other bacteria with similar nutrient preferences as *Olsenella* could a priori be used to restrict VRE intestinal colonization. However, it is important to indicate that *Olsenella* was very efficient at colonizing the gastrointestinal tract upon antibiotic withdrawal and, as the transcriptome analysis pinpointed, it was transcriptionally active in vivo. In addition, different bacteria may have distinct preferences for sugars. The fact that one bacterium is able to consume a particular sugar in vitro does not necessarily mean that it will consume it in vivo if other sugars are available[45,46]. All these considerations should be taken into account before testing alternative probiotics providing similar functions.

We note some limitations to our study: first, we were not able to generate a VRE strain lacking the complete pathway for fructose utilization so that we could characterize the impact of fructose on VRE gut colonization using a mutagenesis approach. Nevertheless, using targeted metabolomics we showed that depletion of fructose by the administration of CBC is significantly associated with a reduction of VRE levels. Moreover, using a customized diet approach we showed that fructose boost VRE growth in vivo. In addition, administration of an excess of fructose restored the capacity of VRE to grow in the presence of CBC both ex vivo and in vivo. Altogether, these results support a mechanism by which CBC-dependent fructose depletion impairs the growth of VRE in the intestinal tract. Second, although we have shown that the administration of a single microbe can restrict VRE intestinal colonization, the complete eradication of VRE was not achieved suggesting that other commensal bacteria, in addition to *Olsenella*, may be required in order to achieve full colonization resistance against VRE colonization. Nevertheless, it is important to mention that both in mice and humans it has been shown that *Enterococcus* dissemination to the bloodstream occurs when this bacterium densely colonized the gut lumen[6,7,47]. Thus the identification of commensal bacteria that can diminish VRE intestinal colonization, as we have done in this study, could be of great value for the development of novel probiotics to prevent VRE bloodstream infections. Moreover, faecal contamination of the hospital environment by VRE-colonized patients specifically occurs in those patients heavily colonized with VRE[9]. Since hospital contamination by multidrug resistant pathogens promotes transmission of these pathogens among patients[11], the identification of commensal bacteria and specific mechanisms that decrease VRE gut levels could promote the development of novel strategies to reduce the propagation of VRE among hospitalized patients.

Besides *Olsenella*, we have previously shown that *Barnesiella*, another isolate initially included in the bacterial cocktail, was associated with protection against VRE intestinal colonization both in humans and mice[20]. This last result was confirmed in the present study. Nevertheless, the specific role that *Barnesiella* plays on colonization resistance against VRE still remains to be elucidated.

Although our study suggests that fructose is key for intestinal colonization of VRE, other sugars may also be relevant for VRE gut colonization and their depletion may be required for a complete eradication of VRE. Indeed, although we detected a significant correlation between the levels of fructose and the capacity of VRE to colonize the intestinal tract, a few mice were colonized with high VRE loads despite the fructose levels were close to those found in untreated mice (Fig. 4b). In addition, VRE could still colonize the murine gut of mice that received a diet that do not contain fructose, although to a lower extent (Fig. 4d). In vitro assays performed in this study have pinpointed other sugars that can be found in the large intestine and that promote in vitro a similar VRE growth. For example, galactose or mannose, sugars that are included in mucins or in other mucosal glycoproteins were efficiently used in vitro by the VRE strain used in this work and by other VRE strains[48]. Future studies will be required to completely decipher the nutritional requirements of VRE in the intestinal tract and to identify the microbiota members that prevent their utilization, both in mice and in patients. Ultimately, all this knowledge could allow the design of personalized microbiome-based therapies to prevent VRE infections. Through metabolomics one could identify the nutrients available for VRE in the gut of a patient, then a proper cocktail of commensal bacteria that efficiently deplete those nutrients could be administered to completely eradicate VRE from the intestinal tract. Although extensive studies will be required to reach this level of knowledge, our study has set up the basis for developing microbiome-based nutrition depletion strategies to prevent infections by these multidrug resistant pathogens.

## Methods

### Animal experiments
All mouse procedures were performed in accordance with institutional protocol guidelines at the "Servei Central de Suport a la Investigació Experimental" at the University of Valencia. Mice were maintained accordingly to the National guidelines (RD 53/2013), under protocols approved by University of Valencia Animal Care Committee describing experiments specific for this study. Experiments were done with 7-week-old C57BL/6J female mice directly purchased from Charles River laboratories (Figs. 1–4a, b and Supplementary Figs. 2–7, 9, 12 and 18) or obtained from the offspring of C57BL/6J mice purchased from Charles River laboratories that were bred in our animal facility (Figs. 4d, f, 5 and Supplementary Figs. 8 and 16). Mice were housed with autoclave-sterilized food (a 1:1 mixture of 2014S Teklad Global diet and 2019S Teklad Global Extrused 19% Protein Rodent Diet from Envigo) and autoclave-sterilized water, except in the experiment shown in Fig. 4d in which mice received the Teklad diet TD05075 for several days as indicated. Temperature was kept at 21 °C +/− 2 °C and humidity was maintained at 60–70%, in 12 h light/dark cycles.

### Animal experiments: Effect of different antibiotics on the microbiota composition and VRE gut colonization
To investigate the effect of different antibiotics on the murine gut microbiota and their relationship with VRE colonization capacity (Fig. 1), groups of 5 mice were treated for 1 week with different antibiotics: vancomycin (0.5 g/l, Alfa Aesar), ampicillin (0.5 g/l, AppliChem) or neomycin (1 g/l, Calbiochem) were given in the drinking water, while ceftriaxone (2.4 mg/ml; 500 μl SC, Sigma) and clindamycin (1.4 mg/ml; 500 μl SC, Fluka) were administered subcutaneously twice daily and ciprofloxacin (0.6 mg/ml; 150 μl IG, Fluka) was administered intragastrically twice daily. As a control, a group of mice ($N = 5$) did not receive any treatment. In this experiment, mice were individually housed from the beginning of the experiment. An overview of the spectrum of action of the different antibiotics used is shown in Supplementary Fig. 1. These antibiotics were selected to target the different fractions of the microbiota (gram positive, gram negative and anaerobes). In the model without recovery, after one week of antibiotic treatment, mice were orally challenged with $10^6$ CFUs of vancomycin-resistant *Enterococcus faecium* strain ATCC700221. This strain was obtained from the ATCC repository. It is resistant to multiple antibiotics including vancomycin and it was isolated from a human faecal sample. This strain has previously been used in other studies of VRE

intestinal colonization using mouse models[7,10,20,49]. A faecal sample was retrieved before the VRE inoculation and conserved at −80 °C to determine the microbiota composition. Two days after VRE inoculation, VRE counts were determined by plating serial dilutions of collected faecal and caecum samples in Gelose BEA plates (Biokar) containing 8 μg/ml of vancomycin (Alfa,Aesar) and 10 μg/ml of ampicillin (AppliChem). We refer to this medium as BEA-AV plates along the present work. Alternatively, in the model with recovery, we assessed VRE colonization capacity after letting the mouse recover their microbiota for two weeks after antibiotic treatment. In this case, mice received the antibiotic treatment for one week, then the antibiotic treatment was stopped to allow the microbiota to recover. Two weeks after antibiotic withdrawal, a faecal sample was retrieved and conserved at −80 °C to determine the microbiota composition. Next, mice were orally challenged with VRE and its levels were assessed two days post-VRE inoculation (p.i.) as previously described.

To check that no bacteria present in the microbiota of the mice (before VRE inoculation) were able to grow in BEA-AV plates and that there was no contamination with VRE previous to its inoculation, faecal samples collected before VRE inoculation were grown on BEA-AV agar plates. As expected, no colonies were detected on BEA-AV plates when faecal samples obtained before VRE inoculation were grown.

### Animal experiments: Test the inhibitory effect of commensal bacteria against VRE intestinal colonization

To assess the capacity of specific bacteria to restore the colonization resistance against VRE, mice were treated with 0.5 g/l of vancomycin in the drinking water for one week and 200 μl of the bacterial inoculum to be tested were administered by oral gavage during 3 consecutive days, starting one day after the antibiotic withdrawal (see below in an additional section the methodology for the preparation of the bacterial inoculum). As control, a group of mice received the buffer used to resuspend the bacteria (i.e. PBS-GC). In order to facilitate the engraftment of the administered bacteria, two mice were co-housed in the same cage after antibiotic withdrawal. Two weeks after the antibiotic withdrawal, a faecal sample was retrieved and conserved at −80 °C to determine the microbiota composition and to check that mice did not contain any bacteria that could grow on BEA-AV plates. Subsequently, mice were housed individually immediately before the VRE inoculation to avoid contamination of VRE between mice from the same cage. Next, mice were orally challenged with 10^6 CFUs of VRE and the level of VRE was assessed as previously described 2 days p.i. In all the experiments we used the VRE strain ATCC700221. In the experiment shown in Supplementary Fig. 8, besides we used the VRE strain AUS0004 and the multidrug resistant *Enterococcus* strain E1162. This mouse model was used for all the experiments of the manuscript in which bacterial isolates were administered, with one exception, the experiment shown in Fig. 5c (therapeutic model). In this particular experiment we wanted to evaluate if *Olsenella* administration, besides restricting VRE intestinal colonization when it was administered before VRE inoculation, could also diminish VRE intestinal levels, once VRE has been first established. This second strategy could be very useful in patients that have already been colonized with VRE. To facilitate the complete establishment of VRE in the gut, we inoculated VRE in the middle of vancomycin administration (after 4 days of treatment), which would enhance VRE intestinal colonization. Subsequently, we maintained vancomycin for another 3 days before stopping antibiotic treatment to facilitate VRE persistence since VRE becomes the dominant bacteria of the gut during antibiotic administration[7]. Briefly, mice were treated with vancomycin (2 mice were housed in the same cage). Four days after initiation of vancomycin treatment, mice were inoculated with 10^6 VRE CFUs and were kept on vancomycin for another 3 days. One day after antibiotic withdrawal, a faecal pellet was retrieved for analysing VRE levels as previously described. Immediately after, the *Olsenella* isolate was administered by oral gavage during 3 consecutive

days as previously described. Two weeks after antibiotic withdrawal, the levels of VRE were measured in faeces. As a control, another group of mice received PBS-GC instead of the *Olsenella* isolate.

### Collection of caecal samples for targeted metabolomics and transcriptomic analysis

To investigate the effect of CBC administration on the caecal transcriptome and fructose levels, mice were treated with vancomycin for one week as described in the previous section. A group of mice treated with vancomycin received CBC during 3 consecutive days, starting the day after antibiotic withdrawal, while the other group of mice received the vehicle PBS-GC instead. Two mice were co-housed in the same cage after antibiotic withdrawal. Two weeks after antibiotic withdrawal, one mouse per cage was sacrificed instead of being inoculated with VRE. The caecal content of each sacrificed mouse was collected and resuspended in twice its volume of phosphate buffer (100 mM Na2HPO4 pH 7.4) respect to weight (considering 100 mg as 100 μl), homogenized and centrifuged (13200 g, 1 min). For targeted metabolomics analysis, the supernatant was separated and frozen using a dry ice bath. For transcriptomic analysis, the pellet was resuspended in 1 ml of RNAlater (Ambion) and kept at 4 °C for 24 h before storage at −80 °C. The other mouse from the same cage was inoculated with VRE and the levels of VRE were analysed 2 days p.i to correlate the capacity of VRE to colonize the gut with the levels of fructose or beta-fructosidases identified in the caecal content of the co-housed mouse (Fig. 4b, Supplementary Fig. 18).

### Evaluation of the similarity in the VRE colonization levels on mice co-housed in the same cage

To evaluate that the capacity of VRE to colonize the mouse from one cage was representative of the capacity of VRE to colonize the other mouse housed in the same cage, mice were treated as described in the previous sections. Briefly, mice were treated for one week with vancomycin and then allowed to recover for 2 weeks after antibiotic withdrawal. A group of mice received CBC by oral gavage during 3 consecutive days, starting one day after antibiotic withdrawal, while the other group of mice received PBS-GC instead. Two mice were housed in the same cage after antibiotic withdrawal. Two weeks after antibiotic cessation, mice were separated into individual cages and were orally challenged with 10^6 CFUs of VRE and the level of VRE was assessed as previously described 2 days p.i.

### Effect of fructose administration on VRE intestinal colonization levels

In order to evaluate the effect of fructose on VRE intestinal colonization levels (Fig. 4f), mice were first treated as described above. Briefly, mice were treated for one week with vancomycin. Subsequently, antibiotic treatment was stopped. A group of pairs of mice received CBC during 3 consecutive days, starting one day after antibiotic withdrawal. While another group of pairs of mice received PBS-GC instead. Two mice were housed in the same cage after stopping antibiotic treatment. Two weeks after antibiotic withdrawal, mice were individually housed. One of the mice from each cage received 15% fructose in the drinking water (a concentration of fructose that was used to overcome the small intestine capacity of fructose absorption and to significantly increase fructose levels in the large intestine, based on experiments performed in a previous work)[50]. As control, the other mouse from the cage received regular water instead. The next day, mice were inoculated with 10^6 CFUs of VRE. Two days after VRE inoculation the levels of VRE were quantified in faecal samples as previously described.

An additional experiment (Fig. 4d) was performed to evaluate the effect of depleting fructose from the diet on VRE intestinal colonization. In this case, mice were treated for one week with vancomycin. Then, pairs of mice were allowed to recover for 2 weeks from vancomycin treatment. Subsequently, one mouse from each cage received the custom Teklad diet TD05075 that does not contain fructose. The

only carbohydrate available in this diet is starch (a polysaccharide formed by glucose monomers). Another group of mice was fed the same type of diet but fructose was supplemented in the drinking water as described above. One day after, mice were inoculated with $10^6$ CFUs of VRE. Two days after VRE inoculation, the levels of VRE were quantified in faecal samples as previously described.

## Commensal bacteria isolation

Since all the bacteria that we wanted to isolate were anaerobic, manipulations were performed inside an anaerobic chamber (Whitley DG250 Anaerobic Workstation, Don Whitley Scientific Limited), supplied with a 10% $CO_2$, 10% $H_2$ and 80% $N_2$ compressed gas mixture (Linde AG®). The material used for bacterial growth (solid or liquid media) was deoxygenated overnight, inside the anaerobic chamber, previous to its use.

With the purpose of growing and isolating intestinal anaerobic commensal bacteria from the mice, we collected the caecum content of 7-week-old SPF-C57BL/6J mice purchased from Charles River laboratories. The caecum content was immediately resuspended in PBS - cysteine 0.1% (approximately the same volume of caecal content as buffer). Subsequently, the tube containing the resuspended caecum content was transported to the laboratory in an anaerobic jar. In the laboratory, working in the anaerobic chamber, 7 ml of PBS-GC were added to the suspension. The suspension was frozen at −80 °C in aliquots of 150 μL. One aliquot was thawed and plated under anaerobic conditions on the culture media Columbia Blood Agar (CBA; VWR) (dilution from $10^{-2}$ to $10^{-6}$). Plates were incubated in the anaerobic chamber for 6 days at 37 °C. Grown colonies were re-streaked in a new CBA plate. Taxonomic identification of each isolate was performed through a colony-PCR of the 16S rRNA gene and Sanger-sequencing of the PCR product. The PCR was performed using the universal primers F27 (ACGAAGCATCAGAGTTTGATCMTGGCTCAG) and R1492 (CGGT TACCTTGTTACGACTT). Each PCR reaction was done in a total volume of 25 μL, using Thermopol® Reaction Buffer 10× (2.5 μL), dNTPs 10 mM (0.625 μL), primer forward 10 μM (0.5 μL), primer reverse 10 μM (0.5 μL), Taq-polymerase (0.5 μL), 1 or 5 μL of bacteria resuspended in PBS as DNA source (according to the concentration of the bacterial suspension) and ultrapure water to adjust the volume to a total of 25 μL. The parameters of PCR reaction were: initial denaturation (5 min, 94 °C); 35 cycles of denaturation (30 s, 94 °C), hybridization (30 s, 56 °C), elongation (30 s, 68 °C). After the 35 cycles, the reaction finalized with an elongation cycle (5 min, 72 °C). PCR products were purified using purification plates ExcelaPure™ 96-well Ultrafiltration plate (Edge-Bio) and were sequenced through capillary electrophoresis on the sequencing facility of the Servicio Central de Soporte a la Investigación Experimental of the University of Valencia. Quality of retrieved sequences in ab.1 format was manually checked using Trev program from package Staden 2.0 (avaible at http://staden.sourceforge.net/) and sequences were kept in fasta format. Phylogenetic classification of the sequences was done using Mothur as described below[51]. The isolated and taxonomically characterized bacteria were stocked at −80 °C in PBS-glycerol 20%.

## Culture of bacterial isolates to test their inhibitory effect against VRE growth in vivo

To preserve the viability of anaerobic bacteria, manipulations were performed inside an anaerobic chamber as described in the previous section. Bacterial isolates were streak from glycerol stocks on CBA plates and were grown during 3 days, except for *Oscillibacter* that was grown for 6 days due to the slower in vitro growth of this isolate. The grown bacterial colonies were resuspended in PBS-GC. These bacterial suspensions were used individually or were combined according to the experiment, but always maintaining an individual absorbance for each isolate of 0.5 ($OD_{600}$), which corresponds to an average of $10^7$ CFUs/ml per isolate. Once prepared, the inoculums were aliquot and frozen at −80 °C. The day of inoculation, the inoculums were placed and

transported on dry ice to the animal facility. Inoculums were thawed immediately before their oral inoculation to mice.

The bacterial isolates that were administered to mice were: *Alistipes* (strain CU970), *Barnesiella*[52], *Olsenella* (CU969), *Oscillibacter* (CU971), *Flavonifractor* (CU972) and *Bacteroides* (strain CU22). All bacterial isolates were obtained in this study except *Barnesiella* that was previously obtained in Memorial Sloan Kettering Cancer Center and kindly provided by Dr. E.G. Pamer[52].

## Ex vivo assays to study VRE inhibition by CBC

As an initial test to investigate if administration of CBC could inhibit VRE growth in the absence of the host (Fig. 2f), caecum contents were collected from mice two weeks after stopping vancomycin treatment. Caecum contents were collected from 2 groups of mice: (i) mice that received CBC during 3 consecutive days, starting the day after stopping vancomycin treatment, (ii) mice that received PBS-GC instead of CBC. Collected caecum contents were placed immediately in an anaerobic jar to keep them on an anaerobic environment until their introduction in an anaerobic chamber. Subsequently, inside the anaerobic chamber, 1250 VRE CFUs (from an overnight culture) were inoculated into 50 mg of caecal contents, resuspended in 100 μl of PBS pH 7. The mixture was incubated during 24 h under anaerobic conditions at 37 °C. Subsequently, dilutions of the mixture were plated on BEA-AV plates to quantify VRE growth.

To test nutrient depletion as a mechanism by which CBC inhibits VRE growth (Figs. 4e, 5a, b), caecum contents were collected from mice two weeks after stopping vancomycin treatment (similar antibiotic treatment as described in previous sections). Caecum samples were frozen at −80 °C until their usage. Caecum samples were thaw and resuspended in PBS pH 7 (100 mg of caecum content in 1 ml of PBS). After resuspending the caecal content, a 2 min centrifugation was performed at 11000 g and the supernatant was collected in a new tube. Subsequently, the supernatant was filtered using centrifugation columns, firstly using columns with membranes of 0.45 μm pore size (Thermo Scientific) followed by a second centrifugation in columns with a membrane of 0.2 μm (Thermo Scientific). 40 μl of the filtered supernatant was added to each well of a 96 well-plate for each condition tested. The plate containing the supernatants was reduced overnight in an anaerobic chamber to remove the oxygen. The next day, stocks of the bacterial isolates (i.e. *Alistipes, Barnesiella, Olsenella, Oscillibacter* and *Flavonifractor*) were unfrozen inside the anaerobic chamber. $10^6$ CFUs of each bacterium were resuspended in a final volume of 5 μl of PBS (containing the 5 bacterial isolates or individual isolates). The 5 μl containing the bacteria were added to the 40 μl of the caecum-filtered supernatants. The mixture was incubated at 37 °C during 24 h. After the incubation period, 5 μl of PBS containing 500 CFUs of VRE (obtained from an overnight culture) were added inside anaerobic chamber to the cultures. As a control, 500 CFUs of VRE were added to caecal filtered supernatants that were previously incubated as described above but that did not contain any commensal bacteria or that contained an isolate of the genus *Bacteroides* (not associated with protection against VRE). Besides the addition of VRE, when specified in the results, an excess of fructose: 5 μl of pre-reduced 0.5 M fructose was added immediately before adding VRE. The sugar was previously resuspended in water and filtered (0.2 μm pore size). As control, pre-reduced filtered water was added instead of fructose. After VRE administration, the mixture was incubated for 6 h under anaerobic conditions at 37 °C. Subsequently, dilutions of the mixture on PBS were plated on the selective media BEA-AV to quantify VRE growth. Controls not containing VRE were also included to verify the absence of colonies on BEA-AV plates when VRE was not inoculated.

## Faecal DNA extraction and biomass calculation

Bacterial DNA from faecal samples was extracted using the QIAamp® DNA Fast Stool Mini kit (QIAGEN, Spain, ref 50951604). Extractions were

performed according to manufacturer instructions with introduction of a previous mechanic disruption step with bead-beating, as we have previously described[53]. Briefly, samples resuspended in the first buffer of the extraction kit were shaken on a Vortex-Genie 2 equipped with a Vortex Adapter (Mobio, ref 13000-V1-24) at maximum speed for 5 min in the presence of 500 μl of glass micro-beads (acid-washed glass beads 150–212 μm, Sigma®). Subsequently we followed the protocol of QIAamp® DNA Fast Stool Mini kit. The purified DNA was eluted in 50 μl of milliQ water and was quantified using Qubit 3.0 Fluorometer. As a measure of microbial community biomass, as previously described[54,55], the nanograms of DNA per gram of faecal sample were calculated.

### 16S rRNA high-throughput sequencing and analysis
The V3-V5 region of the 16S rRNA gene was amplified (Kapa HiFi Hot-Start Ready Mix), indexed with Nextera® XT Index Kit (96 indexes, 384 samples) and sequenced as described in the manual for "16S Metagenomic Sequencing Library Preparation" of the MiSeq platform (Illumina) using Miseq Reagent Kit V3.

Quality assessment of the obtained sequences was performed using printseq-lite v.0.20.4[56]. Sequences were trimmed using the sliding-window technique, such that the minimum average quality score over a window of 20 bases never dropped below 30. Sequences were trimmed from the 3'-end until this criterion was met. Then, trimmed forward and reverse pair-end sequences were assembled using fastq-join v.1.1.2 from ea-utils suite[57], applying default parameters (maximum 8 percent of difference and minimum overlap of 6 bp). Assembled pair-end sequences larger than 400 bp were kept for the subsequent analysis. Sequences were aligned to the 16S rRNA gene using as a template the SILVA reference alignment[58] and the Needleman-Wunsch algorithm with the default scoring options. Potentially chimeric sequences were removed using Uchime v.4.2.40[59]. To minimize the effect of sequencing errors in overestimating microbial diversity[60], rare abundance sequences that differ in 1% from a high abundance sequence were merged to the high abundance sequence using the pre.cluster option in Mothur[51]. Since different number of sequences per sample could lead to a different diversity, when we compared the diversity of different faecal samples, we first rarefied all samples to the number of sequences obtained in the sample with the lowest number of sequences (i.e. 29667 for the analysis of Figs. 1, 2a, Supplementary Figs. 2, 3, 6, and 22904 for the analysis of Fig. 2d, e, Supplementary Fig. 9). Sequences were grouped into OTUs using Vsearch v.2.9.0[61], with the abundance based agc method. Sequences with distance-based similarity of 97% or greater were assigned to the same OTU. Phylogenetic classification was performed for each sequence using the Bayesian classifier algorithm described by Wang and colleagues with the bootstrap cutoff 60%[62]. Classification was assigned to the genus level when possible; otherwise the closest level of classification to the genus level was given, preceded by "unclassified; UC". Shannon index was obtained at the OTU level with Mothur. Richness was defined as the number of OTUs identified in a particular sample. Non-metric multidimensional scaling (NMDS) in two dimensions was applied using function metaMDS and the BrayCurtis distance matrix obtained at the OTU level with Mothur. Singletons (very rare OTUs identified in only one sample, containing only one count) were not included for the calculation of the Shannon index, richness and the BrayCurtis distance.

### Genome sequencing and analysis
The bacteria of interest were cultured in anaerobic conditions on CBA plates. Bacteria were resuspended in PBS and DNA was extracted using QIAamp® DNA Fast Stool Mini kit as previously described. Subsequently, a genomic DNA library was obtained with the kit Nextera® XT DNA Sample Preparation Kit according to manufacturer guide. The samples were indexed with Nextera® XT Index Kit and the genome sequencing was performed with an Illumina MiSeq® System, using Miseq Reagent Kit V3 (paired-end).

An average of 1,114,877 paired-end reads per isolate were obtained. Remaining adaptor sequences were removed from the raw data using Cutadapt v. 1.10[63]. On average, 4% of the reads contained adaptor sequences and were trimmed accordingly. Sequences were then filtered by quality using UrQt v.1.0.18[64]. UrQt trims low quality read extremes to avoid data loss. Nevertheless, to avoid possible misclassification of the short reads, only reads with a size of 75 or higher were further processed. In total, 8.5% of the bases were trimmed, and 1.65% of the reads were removed.

Cleaned genomic data was assembled using SPAdes v. 3.7.1 using the "careful" algorithm to improve the contig reconstruction[65]. A multi-kmer Bruijn graph reconstruction was followed as it has been suggested to improve the assembly outcome. We used 6 different kmer lengths (21, 33, 55, 77, 99, 127), as it is the best kmer combination for estimated read sizes over 250 bp. SPAdes resulted in an average draft genome assembly of 168 contigs with a N50 of 82932 bp (data for every CBC isolate genome is provided in Supp. Data File 4). The average basepair coverage was 33.38X. Open reading frames (ORFs) were identified and annotated using PROKKA v.1.13[66], resulting in an average of 2508.4 ORFs per genome. ORFs were translated into amino-acids and queried against the KEGG database[67]. Annotation was performed using HMMer v.3.1.2[68] with the following parameters: only Hits with an e-value lower than 0.05 and a minimum coverage of 0.50 were kept as significant results. For each ORF, only the best hit was kept.

### Reclassification of the bacterial isolate (Unclassified *Ruminococcaceae*) to *Flavonifractor*
Based on its full 16S rRNA sequence, one of the bacterial isolates that was found to be negatively associated with VRE intestinal colonization (Fig. 2a) was initially classified as Unclassified_*Ruminococcaceae*. To better classify this isolate, we used its genome sequence to perform a taxonomic placement within the family *Ruminococcaceae*, now merged to the family *Oscillospiraceae*.

Unclassified_Ruminococcaceae phylogeny was inferred by constructing a phylogenetic tree based on the core-genome of representative genomes from the family *Ruminococcaceae/Oscillospiraceae*. All the genomes were downloaded from refseq using the program datasets from NCBI (taxID 216572, 21st October 2021), the ones with complete sequence were subsequently used.

The core-genome reconstruction of these genomes and the Unclassified_Ruminococcaceae of interest was done using the software OPSCAN v.0.1 (https://bioinfo.mnhn.fr/abi/public/opscan/)[69]. Briefly, orthologs were identified as bidirectional best hits using an end-gap free global alignment between a reference proteome from the group of interest and each of the other proteomes. Hits with less than 60% amino acid sequence similarity or more than 20% difference in protein length were discarded. The core-genome was defined as the shared group of orthologs genes that were identified in each of the comparison against one of the strains (*Acetivibrio clariflavus* DSM 19732). The core-genome was judged to consist of 83 protein-encoding gene sequences. Then, the proteins encoded by the core-genome were individually aligned using the multi sequence alignment program MAFFT version 7.453 with the -linsi parameter[70]. Non-informative regions of the alignment were trimmed using trimAl v1.4.rev15 with the automated1 algorithm[71]. The resulting alignments were then concatenated for each genome. The phylogenetic tree was inferred using IQ-TREE v.2.0.4[72], and the best model, LG + F + I + G4, was determined using the option TEST. The robustness of the topology was tested with 1000 rapid bootstrap experiments. The phylogenetic tree was visualized with iTol v5.5.1 and duplicated strains were manually deleted[73].

### RNA extraction and metatranscriptome sequencing
Samples conserved in RNAlater were unfrozen on ice and a volume corresponding to 100–250 mg of caecal sample was transferred to a weighted 2.0 ml microtube. Each subsample was diluted with a volume

of ice cold DEPC water equal to its weight and centrifuged for 5 min at 11000 g, 4 °C. The supernatant was discarded and the resulting pellet was weighted. RNA was extracted from the obtained pellets using the kit Power Microbiome RNA isolation kit (reference 26000-50, Mobio) according to manufacturer protocol with slight modifications. First, to improve the process, the first step was performed combining a phenol-chloroform extraction (pH 5) and bead-beating disruption, as suggested in the kit protocol and already documented[74]. Second, to improve the elimination of the DNA, the DNAse treatment step was increased to 45 min at 37 °C instead of 15 min at room temperature. Third, in the final step, the RNA was eluted with 100 µl of PM8 after incubation for 5 min (instead of 1 min). After the extraction, the purity of RNA was checked performing a qPCR with bacterial 16S rRNA universal primer F27 (ACGAAGCATCAGAGTTTGATCMTGGCTCAG) and 338 R (TGCTGCCTCCCGTAGGAGT). If the PCR was positive, it meant that DNA remained in the sample. In case of amplification, the sample was treated with Baseline-ZERO Dnase (Epicentre) to completely remove the remaining DNA. A standard ethanol precipitation protocol was used to precipitate RNA and to wash out the Stop solution that could interfere with the following steps.

Because we were interested in identifying the functions expressed by CBC, we were mainly interested in the mRNA. However, most of the RNA present in a cell is rRNA. To remove the rRNA, every sample was treated with Ribo-Zero rRNA Removal Kit (Bacteria) (Illumina). The library for sequencing was constructed using the ScriptSeq Complete Kit Bacteria according to the manufacturer protocol. Libraries were sequenced using the NextSeq 500/550 High Output Kit v2.5 (150 bp, pair-end) on a Nextseq platform according to the manufacturer protocol.

## Metatranscriptomic analysis

An average of $2.6 \times 10^7$ paired-end reads were obtained for each sample (see in Supplementary Data File 5 number of reads obtained per sample). Adaptor sequences and low-quality reads were removed using Cutadapt v. 1.10 and UrQt v.1.0.18. Although the RNA extraction method used contained a step to remove the ribosomal RNA, this step does not remove the 100% of the ribosomal RNA. For this reason, metatranscriptomic data was mapped, using bowtie v.2.2.9[75], against a Short Ribosomal Subunit database (including 16S and 18S rRNA reference data) from SILVA[76]. Sequences that match with these databases were discarded. The remaining reads were re-mapped against the Long Ribosomal Subunit database from SILVA and the Mouse reference genome v.38 from the NCBI reference repository[77]. Hits mapping any of both databases were discarded from further analysis.

The remaining transcriptomic sequences, representing an average of $7.1 \times 10^6$ reads per sample, were assembled using SPAdes v. 3.7.1 and the "rna" algorithm to improve the transcript reconstruction. A multi-kmer de Bruijn graph reconstruction was followed as it has been suggested to improve the assembly outcome. We used 6 different kmer lengths as described above. ORFs were identified and annotated using MetaGeneMark v.1.0.1[78]. Only ORFs with a minimum size of 50 aa and with at least one extreme complete were kept.

Subsequently, a catalogue of non-redundant (NR) ORFs was constructed. ORFs were clustered at 90% identity using vsearch v2.9.0 with the "cluster smallmem" option[61,79]. NR-ORFs were annotated using KEGG and HMMer as described above. To calculate the average coverage and the total number of mapped reads per ORF, the filtered metatranscriptomic data was mapped against the formatted NR catalogue using bowtie v.2.2.9. Read counts were converted to transcripts per million (TPM), by first dividing the number of read counts by the length of each ORF (RPK). Then we divided the sum of RPK numbers by 1,000,000 (thus resulting in the scaling factor) and finally dividing each RPK by such scaling factor. This normalization results in each sample summing up to 1,000,000 TPMs, making them comparable.

To identify which transcripts were expressed by CBC isolates, cleaned metatranscriptomic data was mapped against the bacterial genomes obtained from the CBC isolates (see section genome sequencing and analysis) using bowtie v.2.2.9 with a percentage of identity greater or equal to 99% and a minimal coverage length of 80%. A similar normalization was performed as previously described (TPM calculated on the total number of reads per sample).

To identify fructanases, we collected the sequences corresponding to fructanases (fructan-beta-fructosidase and levanase) from Uniprot [The UniProt Consortium][80], using only sequences manually curated. Sequences were aligned using MAFFT with the -linsi parameter[70], and manually curated. The resulting curated alignment was used to build a hidden Markov model (HMM) with hmmbuild from HMMer v.3.1.2 using the standard parameters (http://hmmer.org/). Next, this HMM profile was aligned against the RefSeq genome database, using HMMsearch with the -cut_nc filter selected, to extract proteins with a fructanase activity. Only hits with more than 85% coverage and E-value <1e-10 were selected. Since we retrieved a high number of hits annotated as 6-glucose-phosphate kinase (G6PK) based on Prokka annotation, we separated those from the rest. Finally, the metatranscriptomic data was mapped against both sets of sequences using blastn with default parameters. Hits with a coverage higher than 85% and e-value <1e-10 were selected. When the same transcript mapped both models (fructanase and G6PK), only those where the e-value for fructanase was smaller than that of G6PK were considered.

## Fructose detection in caecal contents

Analysis of fructose was performed by gas chromatography–mass spectrometry (GC-MS). Sample treatment consisted in protein precipitation and derivatization to their trimethylsilyl derivatives following a previous work[81]. To do so, 3 volumes of cold methanol were added to 50 µL of sample. After centrifugation, the clean supernatant was collected and vacuum-dried. Dried samples were derivatized by addition of 20 µL of a BSTFA (N,O-bis-(trimethylsilyl)trifluoroacetamide) solution containing 1% TMCS (trimethylchlorosilane) and 10 µL of pyridine and maintained during 3 h at 70 °C. The derivatized extracts were then vacuum-dried and solved in 100 µL of hexane for injection. A standard of fructose was derivatized the same way and used to identify the fructose peak.

GC-MS analysis was performed using an Agilent 7890 A gas chromatograph coupled to an Agilent 7200 accurate mass high resolution GC/Q-ToF. Separation was performed using an Agilent DB-5ms + DG capillary column (30 m × 0.25 mm i.d., 0.25 µm film thickness +10 m Duraguard) using Helium as carrier gas. Mass analysis was operated on EI conditions, recording data in full-scan mode at 70 eV in a mass range of m/z 50 to 600. Ion source, quadrupole and transfer line temperatures were 250 °C, 150 °C, and 290 °C respectively. The integrated peak area of fructose peak at EIC (extracted ion chromatogram) m/z 217.1075 was used to qualitatively determine the monosaccharide.

## Biolog assay for detection of sugars used as carbon sources by VRE and *Olsenella*

To assess the capacity of VRE strain ATCC700221 to grow in the presence of different sugars, we used the AN MicroPlate™ from Biolog. This plate contains in each well a unique carbon source, including the sugars shown in Supplementary Fig. 13. Each well-plate was inoculated with 100 µl of the minimal medium M1 containing VRE. M1 medium consists on 10 g tryptone and 0.5 g yeast extract in 1 L of PBS and has been previously used to define sugars that enable *Enterococcus* growth[82]. Briefly, VRE was grown overnight in a BHI plate at 37 °C in the anaerobic chamber. The M1 medium used in the experiment was previously reduced overnight in the anaerobic chamber. VRE was collected with a swab, resuspended in M1 medium and adjusted to $OD_{600}$ 0.1. The AN MicroPlate™ was inoculated with 100 µl of this mixture in each well. The AN Microplate had to be incubated for 24 h

outside the anaerobic chamber to continuously measure the absorbance. To maintain the anaerobic conditions, 30 µl of autoclaved mineral oil was added on the top of the 96 well-plate. This method maintained the anaerobic conditions as we checked by growing the anaerobic bacterium *Barnesiella*. The lecture of the absorbance was performed in a spectrophotometer Tecan Infinite F200 at 570 nm of wavelength, using Magellan program v.6.6. The plate was incubated at 37 °C. The absorbance was measured every 30 min with previous shaking. The results were analysed subtracting the background of the first lecture. The results are represented in Fig. 4c dividing the absorbance obtained with a specific sugar by the absorbance obtained with the sugar that allowed the maximum growth (i.e. maltotriose). Exactly the same experiment was performed with *Olsenella*, with the exception that in this case the sugar that allowed a higher growth was mannose.

### Generation of VRE mutant strains defective in fructokinases: Selection of the VRE strain and identification of potential fructokinases

First, in order to obtain a VRE strain defective in fructose utilization, we identified in the VRE strain used in the majority of experiments of the manuscript (ATCC700221), genes that could be encoding for fructokinases (including also 1-phosphofructokinases), the first enzymes required for fructose utilization and that are specific for fructose metabolism. We identified a total of 6 genes that putatively encode for fructokinases. We attempt to delete all the 6 putative fructokinases in the same strain, since deletion of one or few genes could be compensated by the expression of the remaining ones. However, we were not able to obtain any transformant of this strain with the shuttle vector (pWS3) used to generate the mutants. For this reason, we used the genetically tractable strain AUS0004[27] (the other VRE strain used in this manuscript) to obtained a strain defective in fructokinases. We first identified putative fructokinases encoded in the AUS0004 strain by analysing the annotated genome in the KEGG database (https://www.genome.jp/entry/efc). Here, we also found a high level of functional redundancy. Specifically, we identified 5 genes that theoretically encode for fructokinases (EFAU004_00683; EFAU004_01835; EFAU004_02555; EFAU004_01659; EFAU004_01953). Three of these genes (EFAU004_00683; EFAU004_01835; EFAU004_02555) encode for a type of fructokinase that metabolize fructose that has been internalize through non-phosphotransferase transport systems (PTS), while another two (EFAU004_01659; EFAU004_01953) encode for 1-phosphofructokinases which metabolize fructose that has been uptake through a PTS system.

### Generation of VRE mutant strains defective in fructokinases: First methodology

To generate the mutants we followed a methodology previously described to replace the genes of interest by a gene conferring resistance to gentamicin: *aac(6′)-aph(2″)* through homologous recombination[83]. Subsequently, the gentamicin marker can be eliminated using Cre recombinase since the gene encoding the marker is flanked by lox sites. Briefly, for each potential fructokinase, (i) the first 50 bp of the gene of interest + the 450 bp upstream (1st PCR) and (ii) the 50 last bp of the gene of interest + the 450 bp downstream (2nd PCR) were amplified using KAPA polymerase and primers described in Supplementary Data File 6. PCRs were performed in a final volume of 25 µl containing 12.5 µl 2X KAPA HiFi Hot Start Ready Mix, 10 µM forward and reverse primers and 10–100 ng of DNA template. The parameters for the PCR reaction were: initial denaturation (3 min, 95 °C); 30 cycles of denaturation (20 s, 98 °C), hybridization (20 s, 60–65 °C), elongation (1 min, 72 °C). After the 30 cycles, the reaction finalized with an elongation cycle (1 min, 72 °C). PCR products were purified using the Nucleo Mag® NGS Clean-up and size select kit, as described in the manufacturer protocol. Note that the primers that hybridize outside the gene contained the restriction enzymes for cloning as specified in

Supplementary Data File 6. The reverse primer from the 1st PCR and the forward primer from the 2nd PCR contain complementary sequences to fused both PCRs and a restriction site for *Avr*II in order to clone the gentamicin resistant cassette between both DNA fragments. Next, a secondary fusion PCR was performed using as DNA template 1 µl of the 2 PCR products obtained (containing flanking regions to the gene of interest, as described above). This PCR was performed using KAPA HiFi Hot Start polymerase as described above and the forward primer from the 1st PCR and reverse primer from the 2nd PCR. These fused PCR products were cloned into the plasmid pWS3 using *Xma*I and *Xho*I restriction sites. Subsequently, the gentamicin resistance gene *acc(6′)-aph(2″)* was amplified from the pZXL5 plasmid[83], with primers described in Supplementary Data File 6 and the KAPA HiFi Hot Start polymerase as described above. Next, the obtained PCR product was purified with the Nucleo Mag® NGS Clean-up and size select kit and cloned using the *Avr*II restriction site inserted within the fused PCR previously cloned in the pWS3 plasmid. The final obtained plasmids (Supplementary Data File 6) were transformed into the strain AUS0004 using a electroporation protocol previously described[82]. Briefly, electrocompetent cells were prepared by diluting 1000 fold an overnight culture in 25 ml of BHI supplemented with 200 mM sucrose and 1% of glycine and grew overnight at 37 °C in a shaking incubator. Next, bacterial cells were centrifuged and resuspended in new pre-warmed 25 ml of BHI supplemented with 200 mM sucrose and 1% glycine and grown for 1 h at 37 °C in a shaking incubator. After incubation, bacterial cells were washed three times with 0.5 M sucrose and 10% ice-cold glycerol (wash buffer) and finally resuspended in 1.2 ml ice-cold wash buffer. 50 µl aliquots of bacterial competent cells were mixed with 10 µl of plasmid (approximately 50 ng/µl of plasmid concentration), transferred into an ice-cooled cuvette (1.5 mm gap) and kept on ice for 30 min. The suspension containing the cells and the plasmid was electroporated using these parameters: a potential of 2.5 kV, capacity of 25 µF and shunt resistance of 200 Ω. Bacterial cells were resuspended in BHI with 0.5 M glucose and incubated for 1 h at 28 °C. The transformants were grown at 28 °C in BHI agar supplemented with gentamicin (300 µg/ml) for 1–3 days. Subsequently, the protocol to achieve double-crossover recombination to obtain the targeted mutants was followed as previously described[83]. One transformant colony was grown overnight in BHI broth supplemented with gentamicin at 28 °C. Next day, the cell culture was diluted 10000 fold in pre-warmed BHI broth without antibiotics and incubated overnight at 37 °C to lose the thermosensitive plasmid and select transformants with the gene conferring gentamicin resistance, inserted in the chromosome by recombination. The culture was plated on BHI agar supplemented with gentamicin and incubated again at 37 °C (the restrictive temperature for plasmid replication). Colonies were re-streaked on BHI agar plates supplemented with spectinomycin (300 µg/ml) and on BHI agar plates containing gentamicin (300 µg/ml). The colonies that were resistant to gentamicin and susceptible to spectinomycin (conferred by the plasmid) were checked by PCR using flanking primers described in Supplementary Data File 6 to confirm the generated deletions and introduction of the gentamicin resistance gene. Using this methodology usually a mutant strain can be obtained after testing 100–200 clones. We were able to obtain a mutant for the fructokinase encoded by the EFAU004_02555 gene. However, after 10 different attempts and more than 1000 clones tested per each fructokinase, we were not able to obtain mutants of the other 4 fructokinases. For this reason, we applied a recently published new methodology[33], based on the expression of the recombinase RecT, in order to obtain mutants in the other genes encoding for potential fructokinases (see next section).

### Generation of VRE mutant strains defective in fructokinases: second methodology

As a second strategy to generate mutants in fructokinases, we applied a novel methodology that could more efficiently replace a gene by a

resistant marker thanks to the overexpression of RecT, a *Enterococcus* phage derived recombinase[33]. Briefly, fragments containing the flanking regions upstream and downstream the gene of interest and the gentamicin resistant marker inserted within both flanking regions were amplified using as DNA template the plasmid constructions previously generated (previous section, Supplementary Data File 6) and the external primers used to generate these plasmid constructions (Supplementary Data File 6). PCR products were purified using Nucleo Mag® NGS Clean-up. PCR products were transformed into VRE strain AUS0004 containing the plasmid pRecT_2[33], which was introduced in this strain as described before for the other plasmids. In order to transform the PCR products, electrocompetent cells of the AUS0004 strain containing the pRecT_2 were obtained as previously described[33]. Briefly, a culture was grown overnight at 37 °C in 10 ml BHI medium supplemented with 2% glycine and 0.5 M sucrose. Next, the culture was centrifuged and the pellet was resuspended in 25 mL of fresh BHI medium containing 2% glycine, 0.5 M sucrose and 1 mM IPTG for RecT induction. Cells were incubated at 37 °C for 1.5 h in a shaking incubator. Next, cells were pelleted by centrifugation at 5000 relative centrifugal force (RCF) for 10 min and resuspended in 1 mL wash buffer-electroporation solution (0.5 M sucrose and 10% ice-cold glycerol) and transferred to a falcon tube. Cells were pelleted at 7000 RCF during 8 min at 4 °C and resuspended in 1 mL of fresh electroporation solution. This step was repeated but the pellet was resuspended in a final volume of 1.2 ml. 50 μL aliquots of cells in electroporation solution were mixed with 3 μg of purified PCR and transferred into an ice-cooled cuvette (1.5-mm gap). The suspension was then electroporated using the following parameters: 2.5 kV, 25 mF and 200 Ω. Cells were then resuspended in 1 mL BHI supplemented with 0.5 M sucrose and transfer to a microtube. Cells were incubated for 2 h at 37 °C without shanking. Cells were plated in BHI agar supplemented with gentamicin (100 μg/mL) at 37 °C for 1–3 days. The mutation introduced was checked in the obtained transformants by PCR using flanking primers described in Supplementary Data File 6. Using this approach we were able to obtain mutant strains for the genes EFAU004_02555, EFAU004_00683 and EFAU004_01659. However, after 3 different attempts, we were not able to obtain any mutant for the genes EFAU004_01835 and EFAU004_01953. Once the mutants were confirmed through PCR, the plasmid pREC_T2 was cured by growing overnight a colony on liquid BHI containing gentamicin (100 μg/ml) at 37 °C in a shaking incubator. Subsequently, the culture was diluted 1:50 in fresh BHI with gentamicin and incubated at 37 °C in a shaking incubator until reaching OD600$_{nm}$ = 0.8. Dilutions of this culture were plated on BHI plates containing gentamicin. Colonies were re-streaked in plates containing spectinomycin (250 μg/ml) to verify the loss of the plasmid (which confers resistance to spectinomycin).

### Study the growth of VRE fructokinase mutants
The WT VRE strain AUS0004 or the mutants for fructokinases genes were grown at 37 °C on minimal media M1 containing 5 mM of either fructose, glucose or mannose. Optical density was monitor as described above during 24 h.

### Statistical analysis
With the objective of identifying significant differences between two groups of samples in the number of CFUs, diversity, richness, biomass, n° of expressed genes, the fructose levels and growth in sugars, a Shapiro-Wilk test of normality was applied to define if the populations under comparison followed a normal distribution. If this was the case, a t-test was applied. If the populations under study did not follow a normal distribution, the non-parametric Wilcoxon rank-sum test was applied. In few occasions, all the groups, except one, followed a normal distribution (Figs. 3b; 4a; Supplementary Fig. 2B and Supplementary Fig. 2C). In this case both tests were applied. Both tests led to similar results but only the t-test results were kept and indicated in the Figure.

The tests applied were two-sided except in the Fig. 5c and Supplementary Fig. 8 in which we applied a one-sided test because the experiment performed was to test the hypothesis that *Olsenella* could recapitulate the CBC inhibitory effect against VRE colonization or that CBC would restrict intestinal colonization of additional VRE strains. In these experiments the only expected result, which we confirmed, was a lower level of VRE CFUs in the group receiving the commensal bacteria as compare to the PBS-GC group.

In order to determine statistically significant differences in the relative abundance of different taxa between groups of samples, the non-parametric Wilcoxon rank-sum test was applied using wilcox.test function in the "stats" R package. A non-parametric test was used due to the nature of metagenomic data (relative abundances), which in most cases violate the main assumption of typical parametric test (normal population in each group), whereas non-parametric tests are much more robust to underlying distribution of the data since they are distribution-free approaches[84]. Only taxa with a minimal signal of detection were included in the statistical analysis (i.e. the feature was present in 50% of the samples from at least one of the two groups under study with an abundance superior to five times the minimum signal detected in all taxa and samples). To adjust for multiple hypothesis testing, we used the FDR approach by Benjamini and Hochberg implemented in the fdr.R package[85].

To identify commensal bacteria associated with protection against VRE, the two-sided spearman correlation test was applied between the relative abundance of the genera identify in a particular mouse in the faecal sample collected before VRE inoculation and the faecal levels of VRE (CFUs/100 mg) in that particular mouse 2 days p.i. To adjust for multiple hypothesis testing, we used the Benjamini and Hochberg approach as described above. Since this analysis was performed to identify commensal bacteria that confer protection against VRE under normal conditions (i.e. untreated mice) and to facilitate the subsequent isolation of the bacteria with a significant association, we focus this analysis on those genera that were more abundant in untreated mice (median abundance in samples collected from untreated mice > 0.4%; Supplementary Fig. 5). All correlations, including low abundant genera and taxa that could not be classified up to the genus level are shown in Supplementary Data File 3.

With the objective of defining if the abundance of the isolated potential protective bacteria was significantly different in mice highly susceptible to colonization as compared to mice less susceptible to colonization, a threshold of 10$^3$ VRE CFUs/100 mg of faeces was utilized to define both groups of mice based on the evidence that patients with VRE faecal levels >10$^3$ CFUs/100 mg are those that can contaminate the hospital environment with VRE and therefore could promote its dissemination[9,11]. Once defined both groups, the two-sided Wilcoxon rank-sum test was applied to study significant differences in the abundance of specific commensals between both groups of samples.

To identify KOs whose expression was increased by CBC administration (Fig. 3c), we first identified those KOs that were overexpressed in the presence of CBC (i.e. a log$_2$FC difference >1 between the group of mice that received CBC after vancomycin treatment as compare to the group of mice that did not received CBC). Since we wanted to identify bacterial functions that could explain resistance to VRE colonization in most mice, we focused on KOs that were prevalent, discarding those that were not present in at least 80% of the samples from one group of mice with an abundance superior to three times the minimum signal detected in all samples. Subsequently, we applied the DeSeq2 algorithm v.1.24.0 with the local fit option[86] to these list of KOs that pass our criteria and adjusted the obtained p values using the FDR approach by Benjamini and Hochberg as described above. Read counts that matched ORFs that could not be annotated using the KEGG database were not included in this analysis.

Pearson correlation analysis was applied to investigate the correlation between the fructose caecal levels and the log$_{10}$ VRE faecal

**Article**

CFUs in co-housed mice. The Pearson test was applied since both variables followed a normal distribution (Shapiro-Wilk test > 0.05). Samples from untreated mice were not included in this analysis. The reason for this is that VRE cannot colonize untreated mice. This is because untreated mice contain an intact microbiota and therefore all the mechanisms that confer colonization resistance, besides fructose depletion. Thus inclusion of these samples in the analysis would not have been informative about the association between fructose levels and the capacity of VRE to colonize the intestinal tract of mice. A similar analysis was performed to analyse the association between the levels of beta-fructosidases and VRE levels.

Results were considered significant when p values were lower than 0.05. In case of multiple hypothesis testing, results were considered significant when q values were lower than 0.05.

### Reasons for the disparity in the number of samples between different groups of mice of the same experiment

In the results shown in Fig. 1, only 3 mice are included in the group that received ceftriaxone (without recovery) because in 2 of the mice the amount of extracted DNA was too low and no 16S rRNA amplification could be achieved.

In the results shown in Fig. 4a, groups of mice (PBS-GC and CBC) initially contained 12 mice. However, one mouse from the group that did not received CBC was excluded from the analysis because after microbiome analysis we detected that this mouse had spontaneously recovered very high levels of *Olsenella* (>10%), the key bacteria in the CBC consortium. Including this mouse in the analysis could interfere with the results considering that the hypothesis to test was that restoration of CBC (specifically *Olsenella*) decreases the levels of fructose which confers resistance to VRE colonization. As expected, the co-housed mouse was highly resistant to colonization ($8 \times 10^2$ CFUs/100 mg). In the results shown in Fig. 4a, one sample from the group that did not received CBC could not be included because of a technical issue with the derivatization of the sugars. In the results shown in Fig. 4a and Supplementary Fig. 9, a lower number of mice were included in the untreated group since we expected a lower variability in these group of mice than in those mice that recovered from vancomycin treatment. In Supplementary Fig. 7 a lower number of mice was included in the group receiving *Bacteroides* due to the limitation in number of mice available for this particular experiment.

### Reporting summary

Further information on research design is available in the Nature Portfolio Reporting Summary linked to this article.

## Data availability

The 16S rRNA sequencing data generated of antibiotic treated mice with and without recovery (corresponding to Figs. 1, 2a, Supplementary Fig. 2, Supplementary Fig. 3, Supplementary Fig. 5, Supplementary Fig. 6, Supplementary Fig. 7) has been deposited in the European Nucleotide Archive (ENA) repository under accession code PRJEB40819 (Supplementary Data File 7). The 16S rRNA sequencing data generated of mice that received CBC or not (Fig. 2d, e, Supplementary Fig. 9) has been deposited in ENA repository under accession code PRJEB40849 (Supplementary Data File 7). The genome sequences data generated has been deposited in the ENA repository under accession code PRJEB40866 (Supplementary Data File 7). The Transcriptomic sequences generated has been deposited in the ENA repository under accession code PRJEB40858 (Supplementary Data File 7). Tables containing the abundance of commensal bacteria identified through 16S rRNA sequencing in mice, the VRE colonization levels of antibiotic treated mice and bacterial gene expression levels of analyzed mice have been included in Supplementary Data File 9.

The following databases were used in this study: RefSeq genome database (https://www.ncbi.nlm.nih.gov/refseq/), SILVA database (https://www.arb-silva.de), PROKKA (https://github.com/tseemann/prokka), KEGG database (https://www.genome.jp/kegg/kegg2.html), Mouse reference genome v.38 from the NCBI reference repository (https://www.ncbi.nlm.nih.gov/genome/52), UNIPROT (https://www.uniprot.org). Source data are provided with this paper.

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

## Acknowledgements

CU was supported by grants from Conselleria d'Innovació, Universitats, Ciència i Societat Digital [AICO/2019/266, CIPROM/2021/053] and grants from the Spanish MINECO/MICINN [SAF2014-60234R, SAF2017-90083-R, PID2020-120292RB-I00/AEI/10.13039/501100011033]. SI and AFD were supported by FPI fellowships from the Spanish MINECO/MICINN. GC was supported by a FPU fellowship from the Spanish MEFP. AQ was supported by a fellowship from FISABIO. We thank the Unidad Analítica (IIS La Fe) for their help with the analysis of fructose levels. We thank the Servicio de Secuenciación y Bioinformática from FISABIO for their help with the high-throughput sequencing. We thank the Servei Central de Suport a la Investigació Experimental for their help with the animal experiments. We thank E.G. Pamer for kindly providing the *Barnesiella* isolate. We thank W. van Schaik for kindly providing the plasmids pZXL5 and pWS3 and H.C. Hang for the plasmid pRecT_2. We thank E.G. Pamer and J.R. Penadés for helpful comments on the manuscript.

## Author contributions

S.I., A.F.D., and C.U. designed research studies. S.I., A.F.D., and C.U. performed mouse experiments. S.I., A.F.D., G.C., and A.Q. performed ex vivo and in vitro studies. S.I., A.F.D., L.P.C., and M.L.N. processed fecal samples to obtain omic data. S.I., M.L.N., L.P.C., A.P.L., M.G.G., and C.U. analyzed omic data. A.F.D., G.C., and A.Q. obtained the mutant strains. S.I. obtained the bacterial isolates. S.I. and C.U. wrote the manuscript. All authors read and approved the final manuscript.

## Competing interests

CU has participated as a consultant of Vedanta Biosciences and The Zambon Group. There is no direct overlap between the current study and these consulting duties. The rest of the authors do not have competing interests.
