## [Peer Review File · Nature Communications]

REVIEWER COMMENTS

Reviewer #1 (Remarks to the Author):

This is an interesting study using a combination of in vivo, ex vivo and in vitro experiments and different “omic” techniques to better understand the role of commensal bacteria in the protection against multidrug resistant pathogens (MRP), specifically vancomycin-resistant Enterococcus (VRE). They report to have identified *Olsenella* sp. as protective taxon against VRE gut colonization through nutrient depletion. The topic is of great interest, however I have concerns that need to be addressed.

Major Comments:

1. How is "protection against VRE intestinal colonization" defined? The authors state that the probac consortium restores protection against VRE colonization following abx treatment in a mouse model. However, according to Figure 2C, VRE colonization is still high ($10^6/10^5$) at day 2 and 7 in the probac group. Restoring protection would imply VRE levels to be below the limit of detection as shown in Figure 1C and D in the no treatment control group. Therefore, according to Figure 1C and D, the probac group does not bring VRE colonization below the limit of detection and the author's claim that the probac consortium is protective is not supported by the evidence presented. The authors could argue that the probac consortium "reduces" colonization levels of VRE, however what is the clinical significance of this reduction? The authors do not comment on the severity of infection in terms of symptoms within this mouse model. If this reduction reduced severity of infection, that could be interesting to readers of this journal. Without those data, the reader is left wondering how this "reduction" in GI counts is relevant to VRE intestinal infection especially considering that other groups have been able to clear intestinal infections of multi drug resistant organisms with fecal transplants (<https://www.ncbi.nlm.nih.gov/pmc/articles/PMC4432040/>).

2. There is insufficient evidence to support the claim that fructose depletion results in "suppression" of VRE. First, in order to show that fructose depletion is important for suppressing growth of VRE, there needs to be evidence that fructose is important for colonization within the mouse model. As shown in supplemental figure 11, VRE can utilize many carbon sources. To support the notion that fructose is vital for colonization, a mutant of VRE without the ability to metabolize fructose can be placed in their mouse model to see if colonization is inhibited. Then a complement of that mutant strain can be placed in their mouse model to see if colonization is restored. Then the complement can be used in their mouse model in the different conditions shown. The authors show a correlation (fig 4b) between fructose levels and cfu, but correlation does not equal causation. At the very least this needs to be discussed as a major limitation.

Other comments:

1) VRE is not only a gut pathogen and is not only an antibiotic associated infection. VRE can colonize the urinary tract and skin wounds in addition to the intestinal tract. The manuscript needs to be edited to reflect this. Also, VRE is not only an antibiotic associated infection, it is also a hospital acquired infection. It can be transmitted through contamination without antibiotic use. This is a problem in the elderly population, since they develop bed sores and can acquire VRE in a hospital setting, for example. This needs to be given as context.

2) The Title needs to be edited, is too vague. It needs to be specified if this is seen in a human or mouse.

3) Multidrug resistant pathogen should be changed to multidrug resistant organism (MDRO)

4) Where were the VRE strains acquired from and how many strains were used? A variety of strains should be used to ensure this isn't strain specific

5) How do you explain the discrepancy between your model suggesting fructose is important for colonization and the Stein-Thoeringer model showing lactose drives expansion of Enterococcus?

Minor comments:

- Have you monitored the intake of the water during the antibiotic period across the different groups?

- Line 303-304: "...in the caecal content of the mice in which we had analysed their transcriptome and in additional mice that followed a similar treatment". According to the description of the Figure 4, fructose levels were analyzed on those mice described in Figure 3 (the ones you analyzed the transcriptome). But, what do you mean by additional mice that followed a similar treatment?

- Line 546: in this line it is explained what you have done to facilitate the engraftment of the administered bacteria (ProBac inoculum). I would suggest adding these lines when you explain the amount and the days you administer the inoculum (Lines 537-539) instead of adding them after the VRE administration. It can be a bit confusing for the reader.

- Almost all the mouse models used in the experiments had the same pattern for antibiotic and VRE administration, except the one used for Figure 4I. Why do you inoculate the VRE just 4 days after the initiation of vancomycin treatment and kept with the antibiotic 3 more days instead of administering it just after the antibiotic treatment (7 days)?

- The explanation of the mouse model in Lines 590-596 is practically the same than the one written on the section above in the same page.

- Linkers and connectors, some of them can be found repeatedly throughout the manuscript. For example, in the statistical analysis section, "In order to..." is written in 5 paragraphs out of 7.

Reviewer #2 (Remarks to the Author):

In this study, Isaac et al. identify commensal bacteria that provide a level of colonization resistance to VRE in the mouse gut. They propose that nutrient competition for fructose is the mechanism underlying CR. I think the study is very interesting and the authors have done a number of complementary experiments to provide support for this conclusion. However, there is only indirect evidence of this, and a conclusive proof would require the use of KO strains of *Enterococcus* and *Olsenella* unable to use fructose. While establishing genetic tools for *Olsenella* is beyond the scope of this study, is it really not possible to generate appropriate KO strains of *Enterococcus*?

With regards to the clinical relevance of this work, would depleting fructose from the diet be sufficient to impair *Enterococcus* colonization? This would seem to be a simpler intervention than going through the process of establishing safety and efficacy of a novel probiotic.

The authors propose that fructose could be available either via overflow of non-absorbed fructose or release from fructans (Lines 411-415). Were fructanases observed in the metatranscriptomic

analysis, and if so was there a correlation between fructanase expression and VRE levels? And which taxa expressed them?

Line-by-line comments

Line 108. Awkward sentence beginning with „In concordance to..“. Please re-phrase.

Line 114. Is sugar availability really an “unidentified mechanism”? Several examples of this were cited in the preceding sentences.

Line 122. Is “unclassified Ruminococaceae” really the most exact classification that can be made. I think it would be informative for the reader if there was a phylogenetic tree showing the position of the isolates in context of other related and characterized strains.

Line 140. In mice?

Line 149. Was there no microbiota analysis of fecal pellets at baseline? If not, then it is not possible to say that the diversity was reduced with antibiotic treatment, but rather that antibiotic-treated mice had lower diversity than controls.

Line 200. Grammar. “Genus level”

Line 208. Grammar. “chose”

Line 221. Grammar. “taxon”

Line 244 (and also 452). “Functionally-active”. “Transcriptionally-active” would be more precise.

Line 284. How many species expressed fructose utilization pathways in non-treated mice? It would be of interest to know if *Olsenella* was the only (or dominant) fructose utilizer, or if other bacteria were also involved in this nutrient niche.

Line 403. Grammar. "...difficults..." is not the correct verb to use here.

Figure 4. Is fructose the only metabolite that was measured? Measuring a single metabolite is not really "metabolomics"...

Reviewer #1 (Remarks to the Author):

This is an interesting study using a combination of in vivo, ex vivo and in vitro experiments and different "omic" techniques to better understand the role of commensal bacteria in the protection against multidrug resistant pathogens (MRP), specifically vancomycin-resistant Enterococcus (VRE). They report to have identified Olsenella sp. as protective taxon against VRE gut colonization through nutrient depletion. The topic is of great interest, however I have concerns that need to be addressed.

We thank the reviewer for her/his interested in our study and for indicating that the topic that we study is of great interest, we have attempted to address all the reviewer concerns as specified below.

Major Comments:

1. How is "protection against VRE intestinal colonization" defined? The authors state that the probac consortium restores protection against VRE colonization following abx treatment in a mouse model. However, according to Figure 2C, VRE colonization is still high ($10^6/10^5$) at day 2 and 7 in the probac group. Restoring protection would imply VRE levels to be below the limit of detection as shown in Figure 1C and D in the no treatment control group. Therefore, according to Figure 1C and D, the probac group does not bring VRE colonization below the limit of detection and the author's claim that the probac consortium is protective is not supported by the evidence presented. The authors could argue that the probac consortium "reduces" colonization levels of VRE, however what is the clinical significance of this reduction? The authors do not comment on the severity of infection in terms of symptoms within this mouse model. If this reduction reduced severity of infection, that could be interesting to readers of this journal. Without those data, the reader is left wondering how this "reduction" in GI counts is relevant to VRE intestinal infection especially considering that other groups have been able to clear intestinal infections of multi drug resistant organisms with fecal transplants (<https://www.ncbi.nlm.nih.gov/pmc/articles/PMC4432040/>).

We agree with the reviewer that the ProBac consortium was not able to completely eliminate VRE in all mice. However, as shown in Fig. 2C, it had a profound effect on VRE intestinal colonization levels. 7 days after VRE inoculation, in 50% of the mice that received ProBac, VRE counts were under the detection limit. Moreover, in those mice in which VRE was not eliminated, VRE levels were 3-4 orders of magnitude lower as compared to mice that did not received ProBac.

Reducing VRE intestinal levels, without completely achieving clearance is clinically relevant since it can decrease the risk of bacteremia and VRE transmission among patients:

- (i) As we and others have shown, high levels of *Enterococcus* intestinal colonization (i.e. when *Enterococcus* dominates the microbiota of hospitalized patients) increase the risk of bacteremia (Ubeda C. et al, J Clin Invest, 2010; Taur Y. et al, Clin Infect Disease, 2013). While patients that are colonized with detectable but lower levels of *Enterococcus* do

not develop bacteremia.

- (ii) Contamination of the hospital environment with VRE and subsequent increase in the risk of transmission occurs when patients are colonized with higher levels of VRE ($>10E3/100\text{mg}$ of feces) (Donskey C. et al, N Engl J Med, 2000, Datta R. et al., Arch Intern Med., 2011). Notably, as shown in Figure 2C, the VRE fecal levels in most mice that received ProBac were below $10E3/100\text{mg}$, while the opposite was observed for those mice that did not receive ProBac.

Thus, identifying members of the microbiome and mechanism involved in restricting VRE intestinal levels, as we have done in this study, could result in novel strategies to decrease the risk of bacteremia and VRE transmission between patients.

We have modified the discussion section in order to emphasize the relevance of restricting VRE gut levels without achieving complete clearance, and therefore, the relevance of identifying new microbiome-based strategies to reduce VRE gut levels. The text was modified as such:

*“Second, although we have shown that the administration of a single microbe can restrict VRE intestinal colonization, the complete eradication of VRE was not achieved suggesting that other commensal bacteria, in addition to *Olsenella*, may be required in order to achieve full colonization resistance against VRE colonization. Nevertheless, it is important to mention that both in mice and humans it has been shown that *Enterococcus* dissemination to the bloodstream occurs when this bacterium densely colonized the gut lumen^{6,7,47}. Thus the identification of commensal bacteria that can diminish VRE intestinal colonization, as we have done in this study, could be of great value for the development of novel probiotics to prevent VRE bloodstream infections. Moreover, fecal contamination of the hospital environment by VRE-colonized patients specifically occurs in those patients heavily colonized with VRE⁹. Since hospital contamination by multidrug resistant pathogens promotes transmission of these pathogens among patients¹¹, the identification of commensal bacteria and specific mechanisms that decrease VRE gut levels could promote the development of novel strategies to reduce the propagation of VRE among hospitalized patients.”*

Nevertheless, in agreement with the reviewer’s comment, we understand that the word “protection” may have not been the most accurate word to describe the phenotype observed in the manuscript and other words such as “restriction”, which we used in the title, may be more appropriate. For this reason and to avoid any confusion regarding the effect of ProBac and *Olsenella* on VRE intestinal colonization, we have modified the text of the manuscript to avoid any sentence/word that could suggest that the commensal bacteria used can completely clear VRE intestinal colonization in all mice. As an example, the consortium, previously called ProBac (Protective Bacterium), has been renamed as Commensal Bacterial Consortium (CBC).

Regarding the question about the symptomatology observed in mice colonized with VRE, it is important to specified that mice do not developed symptoms in the mouse model of VRE intestinal colonization. This is probably due to the fact that VRE remains in the intestinal tract of mice where VRE does not produce any pathology. VRE may

produce symptoms mainly when it disseminates from the gut to the bloodstream. However, several concomitant factors will be required for the development of bacteremia: (i) high levels of VRE intestinal colonization (the aspect studied in the present work), (ii) mucosal injury and immunosuppression, which frequently occur in cancer patients due to the intensive chemotherapy received or (iii) hospital instrumentalization, such as venous catheters, which constitute a point of entrance to the bloodstream. The absence of symptoms in the mouse model used has been indicated now in the results section:

“Despite high levels of VRE intestinal colonization, no signs of pain, distress or discomfort were detected in mice after VRE inoculation.”

Regarding the fecal transplant studies indicated by the reviewer, we agree that some studies have suggested that fecal transplants could be used to decolonize the gut from multidrug-resistant organisms. However, as we already indicated in the discussion, administration of fecal transplants has an inherent risk due to the introduction of multiple bacterial strains in a new host which could result in microbiome-host detrimental interactions. Additionally, the preparation of fecal transplant requires screening for potential pathogens (bacteria, virus and parasites) and cannot be produced in a standardized manner since the composition of the bacterial community will vary accordingly to the donor and sampling time. For this reason, the identification of specific bacteria from the microbiota that restrict VRE colonization can result in a non-antibiotic-treatment that can be more easily applied to the clinic.

2. There is insufficient evidence to support the claim that fructose depletion results in "suppression" of VRE. First, in order to show that fructose depletion is important for suppressing growth of VRE, there needs to be evidence that fructose is important for colonization within the mouse model. As shown in supplemental figure 11, VRE can utilize many carbon sources. To support the notion that fructose is vital for colonization, a mutant of VRE without the ability to metabolize fructose can be placed in their mouse model to see if colonization is inhibited. Then a complement of that mutant strain can be placed in their mouse model to see if colonization is restored. Then the complement can be used in their mouse model in the different conditions shown. The authors show a correlation (fig 4b) between fructose levels and cfu, but correlation does not equal causation. At the very least this needs to be discussed as a major limitation.

We thank both reviewers for their suggestion on generating VRE mutants defective in their ability to utilize fructose. We agree that this could be a direct approach to evaluate the effect of fructose on VRE intestinal colonization. As specified in more detail below, we have attempted to obtain a VRE mutant strain lacking all the genes that encode for fructokinases, the first enzyme involved in the metabolism of fructose, which is essential and specific for fructose utilization. However, as described in the previous version of the manuscript, this was a challenging task due to the high level of genetic redundancy in VRE encoded fructokinases (i.e. 5/6 genes per genome). In fact, after multiple attempts and different applied methodologies (see below), we were not able to disrupt 2 out of the 5 genes encoding fructokinases, suggesting that two fructokinases

may be essential for VRE growth. Nonetheless, we were able to show that the 3 genes that we could mutate are involved in fructose utilization (see a detailed explanation below), confirming that multiple genes in the VRE genome encode for functional fructokinases. This genetic functional redundancy further supports a key role for fructose in VRE metabolism, however, it precludes studying the effect of fructose on VRE gut colonization using a mutagenesis approach without having a proper strain lacking all fructokinases. For this reason, we applied alternative approaches, as described in detail below, based on the manipulation of the levels of fructose and the administration of customize diets that have confirmed *in vivo* that fructose is key for VRE gut colonization and for microbiome dependent restriction of VRE intestinal levels.

First, in order to obtain a VRE strain defective in fructose utilization, we identified in the VRE strain used in the manuscript (ATCC700221), a gene that could be encoding for a fructokinase, the first enzyme required for fructose utilization, and that it is specific for fructose metabolism. Notably, we identified a total of 6 genes that putatively encode for fructokinases, a high level of functional redundancy that suggested that fructose is key in VRE metabolism. We next attempt to delete all the 6 putative fructokinases in the same strain, since deletion of one or few genes could be compensated by the expression of the remaining ones. However, we found that the ATCC70021 could not be genetically modified with the tested approach. We were not able to obtain any transformant of this strain with the shuttle vector required to generate the mutants. For this reason, we tested other VRE strains in order to identify one that could be transformed with the plasmid and that could colonize the murine gut. We identified such strain: AUS0004. This is a clinical VRE strain that was isolated from a bloodstream infection, represents the first *E. faecium* strain whose genome was completely reported and can be genetically modified (Reissier et al., Front. Microbiol. 2021). We demonstrated that this strain is capable of colonizing the murine gut (new **Suppl. Figure 8**). Moreover, we showed that the administration of the bacterial consortium (CBC, previously named ProBac, see previous comment) restricts gut colonization by AUS0004 (new **Suppl. Figure 8**).

Supplementary Figure 8. The commensal bacterial consortium restricts gut colonization of several multidrug-resistant *Enterococcus* strains. (A) Schematic representation of the mouse model used for testing the effect of the bacterial consortium (CBC) on colonization by several multidrug-resistant (MDR) *Enterococcus* strains. Mice were treated during one week with vancomycin. One day after antibiotic cessation, mice received during 3 consecutive days CBC through oral gavage. Two weeks after stopping antibiotic treatment, different strains of multidrug-resistant *Enterococcus*, including two that are resistant to vancomycin (ATCC70021 and AUS0004), were inoculated through oral gavage. Two days post MDR-*E. faecium* inoculation (p.i.), levels of the inoculated strains were detected in faeces. As control, a group of mice received the vehicle for bacteria administration (PBS-GC) instead of CBC. **(B)** Levels of MDR-*E. faecium* strains in faeces. Combined: results from the 3 strains were combined in one graph. The ATCC70021 strain was the strain used in the previous experiments. *** $p < 0.0001$, ** $p < 0.001$, $p < 0.05$, One-sided Wilcoxon rank-sum test.

We next identified putative fructokinases encoded in the AUS0004 strain. Here, we also found a high level of functional redundancy. Specifically, we identified 5 genes that theoretically encode for fructokinases (EFAU004_00683; EFAU004_01835; EFAU004_02555; EFAU004_01659; EFAU004_01953), defined by previous analysis of the genome and comparison with the KEGG database: https://www.genome.jp/dbget-bin/www_bget?gn:efc. Three of these genes (EFAU004_00683; EFAU004_01835; EFAU004_02555) encode for a type of fructokinase that metabolize fructose that has been internalized through non-phosphotransferase transport systems (PTS), while another two (EFAU004_01659; EFAU004_01953) encode for 1-phosphofructokinases which metabolize fructose that has been uptake through a PTS system.

We followed a methodology previously described (Zhang et al; PLoS; 2012) to replace the genes of interest by a gene conferring resistance to gentamycin: *aac(6')-aph(2'')* through homologous recombination. Subsequently, in order to generate double mutants, this resistant marker can be eliminated using a Cre-Lox recombinase approach. In this methodology, the resistant gene flanked by VRE genome fragments are cloned into a thermosensitive plasmid and potential mutants are confirmed by replica plating in selective media. Using this methodology one can usually obtain a mutant strain after testing 100-200 clones. We were able to obtain one mutant for the fructokinase encoded by the EFAU004_02555 gene. However, after 10 different attempts and more than 1000 clones tested per each fructokinase, we were not able to obtain mutants of the other 4 fructokinases. As an alternative approach, we implemented a novel methodology that was published last year (Chen V., et al., Appl Environ Microbiol., 2021) and that could more efficiently replace a gene by a resistant marker thanks to the overexpression of a *Enterococcus* phage derived recombinase. In this approach, a PCR containing the resistant marker flanked by fragments homologous to the VRE genome is transformed in the VRE strain containing a plasmid that overexpress the recombinase. Using this novel approach, we were able to obtain approximately 100 CFUs transformants resistant to gentamycin (theoretical mutants) for the fructokinase encoded by the gene EFAU004_02555 (the one for which we could obtain a mutant following the previous strategy). This number of resistant transformants is similar to the numbers obtained by the previous work implementing the methodology (Chen V., et al., Appl Environ Microbiol., 2021). PCR of 5 selected colonies confirmed that all had the

desired mutation (i.e. replacement of the EFAU004_02555 gene by the gentamycin resistant cassette). In addition, we were able to obtain only one colony resistant to gentamycin when cells were transformed with DNA fragments specific for replacing another 2 fructokinases: EFAU004_00683 or EFAU004_01659. PCR with specific primers of the mutated region confirmed that the two mutants had the desired replacement of the gene of interest by the marker. Since only one mutant could be obtained for these two genes, it is possible that the presence of these genes could be relevant for VRE growth. For this reason, we are sequencing the whole genome of the two obtained mutants to verify that no compensatory mutations were selected, which could have facilitated the growth of these two particular mutants. Notably, after 3 attempts, we were not able to obtain any mutant for the other 2 putative fructokinases (i.e. EFAU004_01835 and EFAU004_01953). This last result indicates that at least in the tested conditions, these 2 genes may be essential for VRE growth, which has precluded our strategy of obtaining a VRE strain that completely lacks fructokinases and therefore the ability of utilizing fructose.

Despite we could not obtain a strain lacking all 5 fructokinases, we evaluated the functionality of the 3 genes that we were able to delete. To this end, we grew the obtained mutant strains in a minimal media containing fructose as carbon source, as previously described in the manuscript. Notably, the 3 mutated genes were involved in fructose utilization by VRE (new Suppl. Fig. 14, see below) but they were not required for the utilization of other sugars frequently found in the gut (i.e. mannose, glucose). This result confirms that all the genes that we could characterize (3 out of 5) are indeed fructokinases. This high level of functional redundancy in the first step of fructose utilization together with the potential essential role for VRE growth of the other two genes encoding fructokinases strongly suggest a key role for fructose and fructokinases in VRE metabolism. We have included these results in the results section and added a new supplementary Figure 14 (see below):

“Next, to further characterize the role of fructose in VRE gut colonization, we pursued a mutagenesis approach in order to obtain a VRE strain not capable of utilizing fructose. To this end, we aimed to delete from the VRE genome all the fructokinases that are involved in the first step of fructose metabolism. We identified 6 genes encoding potential fructokinases (including 1-phosphofruktokinases) in the VRE strain used in the majority of the experiments (ATCC70021). In addition, 5 fructokinases were found to be encoded by the other VRE strain used in this work (Suppl. Fig. 8, AUS0004). This result suggests a high level of functional redundancy for enzymes required for fructose utilization in VRE, which may be indicative of a key role for fructose in VRE growth. Subsequently, we attempt to obtain a VRE mutant lacking all potential fructokinases in the genetically-tractable strain AUS0004²⁷. However, after multiple attempts (see methods), we were not able to disrupt 2 out of the 5 genes encoding potential fructokinases, indicating that, at least in the tested conditions, two of the fructokinases may be essential for VRE growth. Nonetheless, we found that the growth of the 3 mutants that could be generated was reduced in the presence of fructose but not in the presence of other sugars frequently found in the gut (i.e. glucose, mannose, Suppl. Fig. 14), confirming that multiple genes in the VRE genome encode for functional fructokinases. This genetic functional redundancy further supports a key role for fructose in VRE metabolism, however, it precludes studying the effect of fructose on VRE gut

colonization using a mutagenesis approach without having a proper strain lacking all fructokinases.”

Supplementary Figure 14. Deletion of VRE-encoded fructokinases impairs VRE growth using fructose as carbon source. VRE was grown in minimal media M1 supplemented with fructose, glucose or mannose. Cultures were grown for 24 hours and the optical density was monitored every 20 min. The area under the growth curve (AUC) was calculated for the mutants and WT strain. The figure shows the log₂FC difference between the AUCs obtained with a mutant strain and the WT. A negative value indicates lower growth of the mutant as compared to the WT. One-sample t-test compared to 0 (the value representing no differences in growth between WT and mutant strains), ***p<0.001, *p<0.05, ns- not significant. Bargraphs represent the average, whiskers indicate the SEM. The data combines two independent experiments with two replicates in each experiment.

Considering the level of protein redundancy for fructose utilization in VRE, it is likely that deletion of a particular fructokinase may be compensated *in vivo* by the expression of another functional fructokinase. This could preclude reaching robust conclusions regarding the role of fructose on VRE intestinal colonization using a mutant strain in which not all fructokinases have been disrupted. Acknowledging this limitation, which has been included in the discussion section, we pursued an alternative approach that would allow us to demonstrate *in vivo* the relevance of fructose for VRE gut colonization, which, as the reviewer pinpoint, was previously suggested but not confirmed by our correlation statistical analysis of targeted-metabolomic data. To this end, we set up an experiment based on results obtained in a recent published manuscript in which the sugar sorbitol was shown to be required for *Clostridium difficile* gut colonization in mice (Pruss K. et al., Nature, 2021). In this particular experiment, we fed a group of mice with a diet that does not contain fructose (the only carbon source available in this diet is starch, a polysaccharide formed by glucose monomers). Another group of mice was fed the same type of diet but fructose was supplemented in the drinking water, expecting that the presence of fructose would give a growth advantage to VRE in the gut environment. Based on a previously published study (Jang C. et al., Cell Metabolism, 2018), we administered a dose of fructose that would exceed the absorption capacity of the small intestine (i.e. 15% fructose). Thus the administered fructose should reach the large intestine, the site that has been the focus of this work since it is where VRE reach higher levels. Notably, we found that those mice that received fructose in their drinking

water were colonized with significantly higher levels of VRE as compare to those mice that just received the diet not containing fructose (i.e. 3 orders of magnitude higher, new Fig. 4D).

Figure 4. The commensal bacterial consortium restricts VRE intestinal colonization through nutrient competition by depleting fructose, a sugar that boost VRE growth *in vivo*. (A) Caecal fructose levels of the groups of mice described in Fig. 3. a.u.: area units. (B) Pearson correlation analysis between the fructose levels detected in caecum and the faecal log₁₀ VRE CFUs detected in paired co-housed mice 2 days after VRE inoculation. Only samples from Vehicle (circles) and CBC (triangles) groups were included in this analysis (see methods for the explanation about sample inclusion). The line represents the linear regression mean and the grey shadow the 95% CI. (C) VRE growth in Biolog plates using as carbon sources fructose or glucose or without supplementing with an external carbon source. The Y axis indicates the optical density obtained with a particular sugar divided by the optical density obtained with the sugar that allowed a highest VRE growth (i.e. maltotriose, Suppl. Fig. 13). (D) Mice were treated with vancomycin for 7 days and allowed to recover for 2 weeks. One day before VRE inoculation (2 weeks after stopping antibiotic treatment), mice were fed with a diet that do not contain fructose (see methods). Half of the mice received fructose in their drinking water, while the other group of mice received regular water. 2 days after VRE inoculation, the levels of VRE were quantified in fecal pellets. (E) Filtered caecal contents from mice that recovered from vancomycin treatment were inoculated with either PBS, CBC or *Bacteroides* (a bacterium not associated with protection against VRE). 24 h after incubation at 37 °C under anaerobic conditions, VRE was inoculated and the growth was quantified 6 h later. Values represent the change (log₂) in VRE levels after the 6 h. + Fru indicates that immediately before VRE inoculation an excess of fructose was added to the culture. (F) Mice treated with vancomycin were allowed to recover 2 weeks before

VRE inoculation. A group of mice received CBC while the other group received PBS-GC instead. Half mice from each group received fructose in the drinking water, starting one day before VRE inoculation. Levels of VRE were quantified in faeces 2 days after VRE inoculation. * $p < 0.05$, *** $p < 0.001$, **** $p < 0.0001$, ns – nonsignificant, two-sided t-test for A and E; two-sided Wilcoxon rank-sum test for D and F. N=8, 10 and 12 mice per group in (A). N=22 mice in (B). N= 13 mice per group in (D). N=3 biologically independent samples per group in (E). N= 8 mice per group in (F). Bars represent the mean in A and E and the median in D and F.

This result indicates that fructose can support VRE growth *in vivo* and highlights the relevance of dietary fructose in the process of VRE intestinal colonization. We have included these results in the new version of the manuscript as indicated here:

“This genetic functional redundancy further supports a key role for fructose in VRE metabolism, however, it precludes studying the effect of fructose on VRE gut colonization using a mutagenesis approach without having a proper strain lacking all fructokinases. For this reason, we decided to use an alternative strategy to investigate the relevance of fructose in VRE intestinal colonization. To this end, we fed mice with a customized diet that does not contain fructose, or the same diet but supplementing fructose in the drinking water (see methods), expecting that the presence of fructose would give a growth advantage to VRE in the gut environment. Notably, VRE reached significantly higher levels (i.e. 3 orders of magnitude) in those mice that received fructose in their drinking water as compared to mice not receiving fructose (Fig. 4D). This result demonstrated that fructose boosts VRE growth in the intestinal tract.”

To further confirm the relevance of fructose in VRE gut colonization and corroborate *in vivo* that the mechanism by which CBC can restrict VRE gut colonization is by depleting nutrient sources required for VRE growth (i.e. fructose), we performed an additional *in vivo* experiment that we have included in the new version of the manuscript.

In the previous version of the manuscript, we made use of an *ex vivo* assay to show that nutrient depletion by CBC was relevant for inhibiting VRE growth. In addition, we showed that administration of an excess of fructose could boost VRE growth in the presence of the bacterial consortium (Fig. 4E, see above) showing that when there is no limitation for fructose, VRE growth can be restored even in the presence of the bacterial consortium. This result demonstrated, at least *ex vivo*, that the bacterial consortium inhibited VRE growth through depleting nutrient sources such as fructose. To confirm this result *in vivo*, we performed an experiment in which we provided to mice treated with antibiotics and previously colonized with CBC an excess of fructose in the drinking water that could exceed the absorption capacity of the small intestine, as described above. Notably, like in the *ex-vivo* experiment, an excess of fructose restored the growth capacity of VRE to colonize the intestinal tract of mice in the presence of CBC (new Figure 4F, see above). This result further supports that CBC inhibits VRE colonization *in-vivo* through depleting key nutrient sources for VRE growth. This result has been included as a new panel in the figure 4 of the previous version of the manuscript (see above) and it is explained in the results section (see below). Note that part of the Figure 4 (*Olsenella*

experiments) has been moved to a new Figure 5 so that Figure 4 does not become too large. In addition, the previous panel 4D has been eliminated from this version of this manuscript. We removed this panel because based on the study mentioned above (Jang C. et al., Cell Metabolism, 2018), the higher fructose concentration used in the new *in vivo* experiments (15% vs 1.8%) is more adequate to ensure that fructose reach the large intestine, the site where we have monitored VRE and CBC levels. Below we show the text that has been added to results section:

“Previous studies have identified commensal bacteria that can suppress VRE intestinal colonization through production of inhibitory molecules. However, our results suggested that CBC was inhibiting VRE growth through nutrient competition (i.e. fructose depletion). To further confirm this hypothesis, we made use of an ex vivo assay. Briefly, the caecal contents of mice that recovered from vancomycin treatment were filtered and reduced in an anaerobic chamber (see methods). The filtered contents were then incubated in the presence of CBC or a Bacteroides isolate (a non-protective commensal) to allow nutrient depletion. Subsequently, VRE was inoculated and its growth was monitored 6 h after. VRE growth could be detected in filtered intestinal contents that did not contained any bacteria or that had been incubated with Bacteroides (Fig. 4E). In contrast, previous incubation with CBC completely abolished VRE growth (Fig. 4E). To demonstrate that CBC was suppressing VRE growth through depletion of key nutrients (i.e. fructose) and not through production of inhibitory molecules, an excess of fructose, so that fructose would not become a limiting nutrient source, was added to the filtered caecal contents after being incubated with CBC and immediately before VRE inoculation. Notably, addition of fructose was sufficient to restore VRE growth in the presence of CBC (Fig. 4E). Importantly, this result could also be reproduced using the in vivo mouse model. Administration of an excess of fructose in the drinking water to mice that were allowed to recover from vancomycin treatment and were fed with a regular chow diet (see methods) restored the capacity of VRE to colonize the large intestinal tract in the presence of CBC (Fig. 4F). Altogether these results suggest that CBC suppresses VRE growth through depletion of key nutrient sources for VRE growth, in particular, through depletion of fructose, a sugar that promotes the expansion of VRE in the intestinal tract.”

Other comments:

1) VRE is not only a gut pathogen and is not only an antibiotic associated infection. VRE can colonize the urinary tract and skin wounds in addition to the intestinal tract. The manuscript needs to be edited to reflect this. Also, VRE is not only an antibiotic associated infection, it is also a hospital acquired infection. It can be transmitted through contamination without antibiotic use. This is a problem in the elderly population, since they develop bed sores and can acquire VRE in a hospital setting, for example. This needs to be given as context.

We agree with the reviewer that VRE can colonize other parts of the body besides the gut and that VRE is not only an antibiotic associated infection. We have modified the beginning of the introduction to take this into consideration as indicated here:

“VRE can colonize hospitalized patients through contamination of skin wounds or catheters, which can lead to urinary tract infections or bacteraemia⁵. In addition, VRE infections can frequently start by the colonization of the intestinal tract⁶, a crucial step that is suppressed by commensal microbes inhabiting the gut (i.e. the microbiota)⁷.”

2) The Title needs to be edited, is too vague. It needs to be specified if this is seen in a human or mouse.

We have modified the title accordingly. We have indicated that the obtained results were in a mouse model, in addition we have been more specific regarding the bacteria studied in this work. We used now multidrug-resistant *Enterococcus* instead of multidrug-resistant pathogen. Below we indicate the new title:

“Microbiome-mediated fructose depletion restricts murine gut colonization by multidrug-resistant Enterococcus”

3) Multidrug resistant pathogen should be changed to multidrug resistant organism (MDRO)

As mentioned above, we have modified the title accordingly. We have also replaced multidrug-resistant pathogens (MRP) by multidrug-resistant organisms (MDRO) in the abstract.

4) Where were the VRE strains acquired from and how many strains were used? A variety of strains should be used to ensure this isn't strain specific

In our experiments, we only used one strain: the vancomycin-resistant *Enterococcus faecium* ATCC700221 strain. This strain was obtained from the ATCC repository, it is resistant to multiple antibiotics including vancomycin and it was isolated from a fecal sample from a patient. We used this strain because we already used it in previous studies (Ubeda C. et al., J Clin Invest, 2010; Isaac S. et al., J Antimicrobial Chemother, 2017). This information has been included in the manuscript:

“mice were orally challenged with 10⁶ CFUs of vancomycin-resistant Enterococcus faecium strain ATCC700221. This strain was obtained from the ATCC repository. It is resistant to multiple antibiotics including vancomycin and it was isolated from a human fecal sample. This strain has previously been used in other studies of VRE intestinal colonization using mouse models^{7,10,20,50}.”

In order to answer if the inhibitory effect of CBC observed in this study is not strain specific, we made use of other clinical *E. faecium* multidrug resistant strains that we have available in our laboratory: AUS0004, as described above, and E11612, a clinical multidrug-resistant strain that was isolated from the bloodstream of a patient and that

has been used in several mouse models (e.g. Hendrickx AP. et al., mBio, 2015; Heikens E. et al., J Bacteriol, 2007). Notably, we detected that the intestinal colonization capacity of AUS0004 and E1162 was reduced in those mice that had received the bacterial consortium, although the effect was less pronounced for E1162. These results have been included in the manuscript (**Suppl. Fig. 8**, see above):

*“Notably, a similar effect of CBC on gut colonization by resistant enterococci was detected for additional enterococcal strains tested (AUS0004, E1162, **Suppl. Fig. 8**).”*

5) How do you explain the discrepancy between your model suggesting fructose is important for colonization and the Stein-Thoeringer model showing lactose drives expansion of Enterococcus?

We thank the reviewer for mentioning this relevant study, which we already mentioned and discussed in our previous version of the manuscript. We do not think that there is a discrepancy between our study and the study performed by Stein-Thoeringer et al. In their study they showed that lactose is a nutrient source that is relevant for *Enterococcus* expansion. However, depletion of lactose from the diet led to a \cong 10-fold reduction in Enterococcal levels but not to a complete depletion of *Enterococcus*. Thus, other carbon sources, besides lactose, may be supporting VRE growth in the intestinal tract. In our study we have identified an additional carbon source that is relevant for VRE growth *in vivo* (i.e. fructose). Nevertheless, as we indicated in the manuscript and taking into account the study mentioned by the reviewer, we attempted to quantify the levels of lactose in the caecal content from mice in order to investigate if CBC could also be contributing to VRE inhibition through lactose depletion. However, we were not able to detect lactose in the caecal contents of the mice from our study (not shown). This result suggests that either lactose is depleted by commensals that recovered after vancomycin treatment (independent of CBC administration) or that most lactose was digested and absorbed by epithelial cells in the small intestine. Thus, in our mouse model, lactose depletion does not seem to be contributing to the inhibitory effect of CBC on VRE colonization of the large intestinal tract.

We have included this information in the discussion section:

“We attempted to quantify the levels of lactose in the caecal content from mice to investigate if CBC could also be contributing to VRE inhibition through lactose depletion. However, we were not able to detect lactose in the caecal contents of the mice from our study (not shown). This result suggests that either lactose is depleted by commensals that recovered after vancomycin treatment (independent of CBC administration) or that most lactose was digested and absorbed by epithelial cells in the small intestine. Thus, in our mouse model, lactose depletion does not seem to be involved in the CBC mediated inhibition of VRE growth in the large intestine.”

Minor comments:

- Have you monitored the intake of the water during the antibiotic period across the different groups?

We have not monitored the water intake during the experiments. However, as the treatment last for seven days and the mice did not show any sign of discomfort or dehydration, we can be confident that they drank during this period and therefore received antibiotics. Note that with this experiment we did not attempt to comprehensively study the effect of antibiotics on the microbiota composition, which has been characterized in previous studies. Rather, we used antibiotic therapy as a tool to disturb the microbiota composition and correlate the microbiota changes with the capacity of VRE to colonize the intestinal tract. Using this tool, we were able to identify commensal bacteria associated with resistance against VRE colonization.

- Line 303-304: “...in the caecal content of the mice in which we had analysed their transcriptome and in additional mice that followed a similar treatment”. According to the description of the Figure 4, fructose levels were analyzed on those mice described in Figure 3 (the ones you analyzed the transcriptome). But, what do you mean by additional mice that followed a similar treatment?

For transcriptomic analysis we analyzed 6 mice per group (i.e. untreated mice, mice that recovered from vancomycin treatment and mice that recovered from vancomycin treatment that received CBC). When we were going to perform the analysis of fructose levels, based on the suggestions of the experts on metabolomics that have participated in this study, we decided to increase the number of samples to analyze considering the variability in the levels of metabolites that usually is detected in intestinal samples. For this reason, we analyze through GC-MS the same samples described in Figure 3 and additional samples obtained from additional mice that followed the same type of treatment (i.e. additional untreated mice, additional vancomycin-treated mice and additional vancomycin-treated mice that received the bacterial consortium). We have modified the text accordingly to make this clearer:

“To test this hypothesis, we first check the levels of fructose in the caecal content of the mice in which we had analysed their transcriptome and in additional mice that followed a similar treatment (i.e. additional untreated mice, mice that recovered from vancomycin treatment that received either CBC or PBS-GC, Fig. 4A).”.

- Line 546: in this line it is explained what you have done to facilitate the engraftment of the administered bacteria (ProBac inoculum). I would suggest adding these lines when you explain the amount and the days you administer the inoculum (Lines 537-539) instead of adding them after the VRE administration. It can be a bit confusing for the reader.

We agree that the way it was written the text was a bit confusing. We have modified the text according with the reviewer’s suggestion.

“To assess the capacity of specific bacteria to restore the colonization resistance against VRE, mice were treated with 0.5 g/l of vancomycin in the drinking water for one week

and 200 μ l of the bacterial inoculum to be tested were administered by oral gavage during 3 consecutive days, starting one day after the antibiotic withdrawal (see below in an additional section the methodology for the preparation of the bacterial inoculum). As control, a group of mice received the buffer used to resuspend the bacteria (i.e. PBS-GC). In order to facilitate the engraftment of the administered bacteria, two mice were co-housed in the same cage after antibiotic withdrawal. Two weeks after the antibiotic withdrawal, a faecal sample was retrieved and conserved at -80°C to determine the microbiota composition and to check that mice did not contain any bacteria that could grow on BEA-AV plates. Subsequently, mice were housed individually immediately before the VRE inoculation to avoid contamination of VRE between mice from the same cage. Next, mice were orally challenged with 10^6 CFUs of VRE and the level of VRE was assessed as previously described 2 days p.i.”.

- Almost all the mouse models used in the experiments had the same pattern for antibiotic and VRE administration, except the one used for Figure 4I. Why do you inoculate the VRE just 4 days after the initiation of vancomycin treatment and kept with the antibiotic 3 more days instead of administering it just after the antibiotic treatment (7 days)?

In this particular experiment we wanted to evaluate if *Olsenella* administration, besides restricting VRE intestinal colonization when it was administered before VRE inoculation, could also diminish VRE intestinal levels once VRE has been first established. This second strategy could be very helpful for patients that have already been colonized with VRE. To facilitate the complete establishment of VRE in the gut, we inoculated VRE in the middle of vancomycin administration, which theoretically would allow the highest levels of VRE colonization. Subsequently, we maintained vancomycin for another 3 days before stopping antibiotic treatment to facilitate VRE persistence since VRE becomes the dominant bacteria of the gut during antibiotic administration (Ubeda et al., J Clin Invest, 2010). This will also mimic the conditions of hospitalized patients in which the patient is colonized during antibiotic administration. Even in this scenario in which VRE has a clear advantage over the incoming *Olsenella*, the administered commensal was able to significantly reduce VRE levels. This result suggest that *Olsenella* could potentially be used as a therapeutic agent to reduce levels of VRE in colonized patients.

We have explained with more detail the rationale of this second model in the methodology section:

“In this particular experiment we wanted to evaluate if Olsenella administration, besides restricting VRE intestinal colonization when it was administered before VRE inoculation, could also diminish VRE intestinal levels, once VRE has been first established. This second strategy could be very useful in patients that have already been colonized with VRE. To facilitate the complete establishment of VRE in the gut, we inoculated VRE in the middle of vancomycin administration (after 4 days of treatment), which would enhance VRE intestinal colonization. Subsequently, we maintained vancomycin for another 3 days before stopping antibiotic treatment to facilitate VRE persistence since VRE becomes the dominant bacteria of the gut during antibiotic administration⁷.”

Note that now the *Olsenella* results and this model of infection has been included in the new Figure 5, see below. In which we have added a scheme to better show the differences between both models and the purpose of the models (preventive vs therapeutic strategy).

Figure 5. *Olsenella* recapitulates the inhibitory effect of the bacterial consortium (CBC) against VRE colonization. (A-B) Filtered caecal contents from mice that recovered from vancomycin treatment were inoculated with either PBS, CBC or individual isolates as depicted. 24 h after incubation at 37 °C under anaerobic conditions, VRE was inoculated and the growth was quantified 6 h later. Values represent the change (log₂) in VRE levels after the 6 h. + Fru indicates that immediately before VRE inoculation an excess of fructose was added to the culture. **(C)** Prevention model: mice were treated with vancomycin and one day after stopping antibiotic treatment received either *Olsenella* or PBS-GC for 3 consecutive days. Two weeks after antibiotic withdrawal mice were inoculated through oral gavage with VRE and the VRE levels were quantified 2 days later. Therapeutic model: mice were treated with vancomycin for 4 days and inoculated with VRE. Mice were maintained on vancomycin for 3 more days. One day after stopping antibiotic treatment mice received *Olsenella* for 3 consecutive days. Two weeks after stopping antibiotic treatment VRE levels were measured in faeces. *p<0.05, **p<0.01, ****p<0.0001, ns – non-significant, two-sided t-test for A and B. One-sided Wilcoxon rank-sum test for C. N=3 biologically independent samples in A-B. N= 7 mice per group in the prevention model and N=9 mice per group in the therapeutic model in C. Bars represent the mean in A-B and the median in C.

- The explanation of the mouse model in Lines 590-596 is practically the same than the one written on the section above in the same page.

We have modified this part of the text since we have applied two new in vivo models to evaluate the effect of fructose administration on the VRE gut levels.

- Linkers and connectors, some of them can be found repeatedly throughout the manuscript. For example, in the statistical analysis section, "In order to..." is written in 5 paragraphs out of 7.

We agree with the reviewer that we have overuse some connectors, we have modified the text accordingly.

Reviewer #2 (Remarks to the Author):

In this study, Isaac et al. identify commensal bacteria that provide a level of colonization resistance to VRE in the mouse gut. They propose that nutrient competition for fructose is the mechanism underlying CR. I think the study is very interesting and the authors have done a number of complementary experiments to provide support for this conclusion. However, there is only indirect evidence of this, and a conclusive proof would require the use of KO strains of Enterococcus and Olsenella unable to use fructose. While establishing genetic tools for Olsenella is beyond the scope of this study, is it really not possible to generate appropriate KO strains of Enterococcus?

We thank the reviewer for considering our study very interesting and for indicating that we have performed several complementary experiments to support our conclusion. We agree with the reviewer that generating *Enterococcus* mutants that are not capable of using fructose is a direct approach to evaluate the effect of fructose on VRE intestinal colonization. In fact, the same strategy was proposed by the first reviewer in her/his second comment. As replied in detail to the first reviewer, we have attempted to obtain a VRE mutant strain lacking all the genes that encode for fructokinases, the first enzyme involved in the metabolism of fructose, which is essential and specific for fructose utilization. However, after applying different methodologies, we were not able to obtain such VRE mutant strain. For this reason, we have used alternative approaches based on the manipulation of the levels of fructose that have demonstrated that fructose is relevant for VRE gut colonization and for microbiome-dependent restriction of VRE intestinal levels (see above the reply to the second comment of the first reviewer).

With regards to the clinical relevance of this work, would depleting fructose from the diet be sufficient to impair Enterococcus colonization? This would seem to be a simpler intervention than going through the process of establishing safety and efficacy of a novel probiotic.

We thank the reviewer for this great suggestion. As we have indicated in the reply to the previous question (see reply to the second comment of reviewer one), VRE is still capable of colonizing the gut when mice received a diet that did not contain fructose, which suggests that besides fructose, depletion of other sugars may be required to completely clear VRE. Nevertheless, using this approach we were able to

show that supplementation of fructose in the drinking water significantly increase VRE intestinal colonization (i.e. 3 orders of magnitude) suggesting a major role for dietary fructose in gut colonization. This last result also suggests, in line with the reviewer comment, that limitation of fructose in the diet may be a potential strategy to restrict VRE gut colonization.

The authors propose that fructose could be available either via overflow of non-absorbed fructose or release from fructans (Lines 411-415). Were fructanases observed in the metatranscriptomic analysis, and if so was there a correlation between fructanase expression and VRE levels? And which taxa expressed them?

We thank the reviewer for the suggested analysis. We were able to detect expression of fructanases on caecal samples (see new addition in the methodology section). We detected expression of levanase, which can hydrolyze levan; and beta-fructosidase, a broader spectrum fructanase which can hydrolyze levan, inulin and sucrose. We did not find any association between the levels of levanase transcripts and the fecal levels of VRE ($p > 0.3$). In contrast, we did find a positive association, albeit not statistically significant ($p = 0.079$), between the expression levels of beta-fructosidases and the fecal VRE levels (new **Suppl. Fig. 17**). Interestingly, this association was detected in samples from vancomycin-treated mice (**Suppl. Fig. 17A**; $r = 0.76$, $p = 0.079$), but it was lost in caecal samples from vancomycin-treated mice that had received the bacterial consortium (CBC) (**Suppl. Fig. 17B**; $r = -0.02$; $p = 0.95$). Since the expression levels of beta-fructosidases did not differ between both groups of mice ($p > 0.99$), this lack of association in mice that received the bacterial consortium may be explained by the consumption of available fructose by the bacterial consortium. Nevertheless, further studies that exceed the scope of this manuscript should be performed in order to validate this hypothesis. We have included these new analysis as a supplementary Figure and describe them in the discussion section where we mentioned the potential role of fructanases in increasing the available fructose for VRE growth. Below we show the new Supplementary Figure and text added to the discussion section.

Supplementary Figure 17. Association between the levels of beta-fructosidase and VRE levels. Spearman correlation between the caecal levels of beta-fructosidases

expressed by the microbiome and the VRE faecal levels in cohoused colonized mice 2 days after VRE inoculation (see methods). Expression of beta-fructosidases was analyzed 2 weeks after stopping the administration of vancomycin treatment of mice that received the bacterial consortium (CBC) after cessation of the antibiotic treatment **(B)** or received the bacterial vehicle instead **(A)**. P and r values were calculated using the Pearson test. VRE CFUs are in log₁₀ scale so that both variables would follow a normal distribution. N=6 mice.

“There, fructans can be hydrolysed into fructose by extracellular bacterial enzymes (fructanases, including beta-fructosidases and levanases) and the liberated fructose is then internalized through specific transporters to serve as a nutrient source³⁹. Therefore, commensals encoding fructanases could be promoting VRE colonization. In line with this hypothesis, preliminary analysis of the transcriptomic data detected a positive association, albeit not statistically significant, between the expression levels of beta-fructosidases (a type of fructanase) and the fecal VRE levels (Suppl. Fig. 17). Interestingly, this association was detected in caecal samples from vancomycin-treated mice (Suppl. Fig. 17A; $r=0.76$, $p=0.079$), but it was lost in samples from vancomycin-treated mice that had received CBC (Suppl. Fig. 17B; $r=-0.02$; $p=0.95$). Since the expression levels of beta-fructosidases did not differ between both groups of mice ($p>0.99$), this lack of association in mice that received CBC might be explained by the consumption of available fructose by the bacterial consortium. Nevertheless, we acknowledge that this analysis was performed with a limited number of mice (N=6) and that further studies should be performed in order to validate this hypothesis and identify potential commensal bacteria from the microbiome that may enhance VRE intestinal colonization through liberation of nutrients, including fructose.”

It would be very interesting, as the reviewer suggested, to identify the commensal bacteria that encode for the detected fructosidases. However, this is not possible with the available data since the short reads from the transcriptomic analysis does not allow the taxonomic identification of the commensal bacteria from the murine gut that express those fructosidases. To this end, assembly of metagenomes from the same samples would be required, in addition to the identification of metagenomic species (MAGs) within the metagenomes and subsequent mapping of the transcriptome reads against the identified MAGs. Moreover, upon identification, it would be necessary to validate the obtained results by performing *in vitro* and *in vivo* assays to confirm that the identified species can increase fructose availability and VRE growth through the expression of fructosidases. Thus, additional extensive work that we think exceeds the scope of this manuscript would be required to identify the species that may promote VRE colonization through fructosidase expression.

Line-by-line comments

Line 108. Awkward sentence beginning with „In concordance to..”. Please re-phrase.

We have re-phrased this sentence to make it more clear:

“Consistent with a major effect of sugar availability on *Enterococcus* gut colonization,...”

Line 114. Is sugar availability really an “unidentified mechanism”? Several examples of this were cited in the preceding sentences.

In the preceding sentences we indicated that sugar availability seems to be key for intestinal colonization by *Enterococcus*. However, in the mentioned examples, the role of the microbiota in limiting sugar availability was not studied. This is why we indicated in the manuscript that depletion of sugars by commensals could be an unidentified mechanism by which the microbiota confers protection against VRE colonization.

Line 122. Is “unclassified Ruminococcaceae” really the most exact classification that can be made. I think it would be informative for the reader if there was a phylogenetic tree showing the position of the isolates in context of other related and characterized strains.

We thank the reviewer for this suggestion. We have performed a phylogenetic tree including all species from the family Ruminococcaceae, now merged to the family *Oscillospiraceae*. We found that our isolate shares the same branches of the tree as species from the genera *Flavonifractor*. We have decided to include this information in the manuscript as a new Supplementary Figure and refer to the obtained isolate as a strain of *Flavonifractor* rather than unclassified Ruminococcaceae.

Supplementary Figure 10. The unclassified *Ruminococcaceae* isolate is included within the clade of the genus *Flavonifractor*. Phylogenetic tree was constructed with the core-genome of 52 reference genomes of the families *Ruminococcaceae/Oscillospiraceae* plus the isolate of interest (in bold font).

Line 140. In mice?

This sentence has been modified accordingly.

Line 149. Was there no microbiota analysis of fecal pellets at baseline? If not, then it is not possible to say that the diversity was reduced with antibiotic treatment, but rather that antibiotic-treated mice had lower diversity than controls.

We agree with the reviewer that we did not explained these results accurately. We have modified the sentence accordingly:

“All mice that received antibiotics had lower microbiota diversity as compared to untreated mice (Suppl. Fig. 2A), although the effect was greater for those mice that received vancomycin and clindamycin and non-significant for those that received ciprofloxacin. Similarly, a lower microbiota richness (i.e. number of identified Operational Taxonomical Units – OTUs) and faecal biomass (ng of DNA/g of faeces) was detected in those mice that received antibiotics as compared to untreated mice (Suppl. Fig. 2B-C).”

Line 200. Grammar. “Genus level”

Thanks for noticing this mistake. We have written “Genus” instead of “Genera”

Line 208. Grammar. “chose”

Thanks for noticing this mistake. We have written “chose” instead of “choose”

Line 221. Grammar. “taxon”

Thanks for noticing this mistake. We have written “taxon” instead of “taxa”

Line 244 (and also 452). “Functionally-active”. “Transcriptionally-active” would be more precise.

We agree that transcriptionally would be more precise. We have written

“Transcriptionally active” instead of “Functionally active”

Line 284. How many species expressed fructose utilization pathways in non-treated mice? It would be of interest to know if *Olsenella* was the only (or dominant) fructose utilizer, or if other bacteria were also involved in this nutrient niche.

As indicated above, identify other commensal bacteria that encode fructose utilization pathways is not possible with the available data since the short reads from the transcriptomic analysis does not allow the identification of the commensal bacteria from the murine gut that express genes related to fructose metabolism. To this end, assembly of metagenomes from the same samples would be required, in addition to the identification of metagenomic species (MAGs) within the metagenomes and subsequent mapping of the transcriptome reads against the identified MAGs.

We were able to map sequences against the available genomes of the commensal bacteria isolated from the consortium which allow us to identify *Olsenella* as the commensal bacterium, within the consortium, that mainly express the identified fructose transporter (K02770). Note that expression of this transporter could also be detected (although to a lower extent) in mice not having *Olsenella* (i.e. vancomycin treated mice). Thus other commensal bacteria could be expressing this transporter and potentially other enzymes related to fructose metabolism. However, although we appreciate the reviewers comment and it will be very interesting to identify other bacteria capable of utilizing fructose, we cannot perform this analysis since we do not have the genome sequences of other bacterial species present in the mice used in our study.

Line 403. Grammar. “...difficults...” is not the correct verb to use here.

We have now written “hinders” instead of “difficults”

Figure 4. Is fructose the only metabolite that was measured? Measuring a single metabolite is not really “metabolomics”...

We agree that fructose was the only metabolite that has been studied. However, to measure the levels of fructose in the gut we made use of targeted-metabolomic techniques. Thus to be more accurate, we have replaced the word metabolomics with the words “targeted-metabolomics” in all the manuscript.

REVIEWERS' COMMENTS

Reviewer #1 (Remarks to the Author):

You have thoughtfully addressed my critiques.